# Sexual dimorphism in obesity is governed by RELMα regulation of adipose macrophages and eosinophils

**Jiang Li[1†], Rebecca E Ruggiero-Ruff[1†], Yuxin He[1], Xinru Qiu[2], Nancy Lainez[1], Pedro Villa[1], Adam Godzik[1], Djurdjica Coss[1]\*, Meera G Nair[1]\***

[1]Division of Biomedical Sciences, School of Medicine, University of California Riverside, Riverside, United States; [2]Graduate Program in Genetics, Genomics and Bioinformatics, University of California Riverside, Riverside, United States

**\*For correspondence:**
Djurdjica.Coss@medsch.ucr.edu (DC);
meera.nair@medsch.ucr.edu (MGN)

[†]These authors contributed equally to this work

**Competing interest:** The authors declare that no competing interests exist.

**Abstract** Obesity incidence is increasing worldwide with the urgent need to identify new therapeutics. Sex differences in immune cell activation drive obesity-mediated pathologies where males are more susceptible to obesity comorbidities and exacerbated inflammation. Here, we demonstrate that the macrophage-secreted protein RELMα critically protects females against high-fat diet (HFD)-induced obesity. Compared to male mice, serum RELMα levels were higher in both control and HFD-fed females and correlated with frequency of adipose macrophages and eosinophils. RELMα-deficient females gained more weight and had proinflammatory macrophage accumulation and eosinophil loss in the adipose stromal vascular fraction (SVF), while RELMα treatment or eosinophil transfer rescued this phenotype. Single-cell RNA-sequencing of the adipose SVF was performed and identified sex and RELMα-dependent changes. Genes involved in oxygen sensing and iron homeostasis, including hemoglobin and lncRNA Gm47283/Gm21887, correlated with increased obesity, while eosinophil chemotaxis and response to amyloid-beta were protective. Monocyte-to-macrophage transition was also dysregulated in RELMα-deficient animals. Collectively, these studies implicate a RELMα–macrophage–eosinophil axis in sex-specific protection against obesity and uncover new therapeutic targets for obesity.

## Editor's evaluation

In this study, Li and al describe valuable insights into mechanisms underlying sex-differences for diet-induced obesity in mice, demonstrating a role for macrophage-derived RELMa secretion in determining female-specific obesity protection. They provide evidence for the impact of RELMa signaling in eosinophil recruitment for diet-induced obesity protection in female mice. Single-cell RNA-seq analysis of the stromal vascular fraction of control and RELMa deficient animals were used to investigate molecular mechanisms underlying the protection. The conclusion of these findings provides evidence supporting dysregulation of cell differentiation pathways.

## Introduction

Obesity is an epidemic of significant public concern and contributes to the increased risk of several diseases, including type 2 diabetes, cardiovascular disease, nonalcoholic fatty liver disease, and COVID-19. Currently in the US, over 30% of men and women are classified as obese, with a body mass index (BMI) of ≥30 kg/m² (*The Lancet Public Health, 2018*). There are profound sex differences in adipose tissue deposition and obesity-associated diseases (*Link and Reue, 2017*). Obese men are more at risk for metabolic syndrome, cardiovascular disease, and myocardial infarction than obese

women (*Gerdts and Regitz-Zagrosek, 2019*). Male mice fed high-fat diet (HFD) gain more weight and have an increased risk of insulin resistance than females (*Parks et al., 2015*). Despite these sex differences, most studies have historically focused on obesity mechanisms in males, since males gain weight more rapidly than females (*Camporez et al., 2019*). Therefore, there remain many gaps in knowledge about the underlying mechanisms for obesity and whether these are sex dependent, which can impact the development of therapeutics that are equally effective for both males and females. To address this gap, the focus of recent studies has been identifying mechanisms that provide protection in females (*Chen et al., 2021a*; *Lainez et al., 2018*). Males and females accumulate fat into different adipose tissue depots; males deposit more fat into visceral adipose depots, while females deposit fat preferentially into subcutaneous depots (*Lainez et al., 2018*; *Palmer and Clegg, 2015*). Since visceral adiposity is associated with the metabolic syndrome (*Wajchenberg, 2000*), differential fat accumulation may explain male propensity for obesity-mediated pathologies.

An underlying immune component for obesity pathogenesis is well recognized, with obesity being regarded as a chronic inflammatory process. Macrophages are critical immune effectors in obesity. Increases in adipose tissue size correlate with macrophage infiltration into the fat depots and proinflammatory cytokine production in both humans and mice (*Weisberg et al., 2003*; *Curat et al., 2006*; *Olefsky and Glass, 2010*). Obese adipose tissues produce increased levels of leptin (*Chen et al., 2021a*; *Gruen et al., 2007*), and monocyte chemoattractant protein-1 (MCP-1, or CCL2 chemokine) that binds CCR2 (*Kanda et al., 2006*; *Kaplan et al., 2015*), which may serve as chemoattractant to recruit monocytes. In turn, they can be activated or differentiate into macrophages, initiating the secretion of cytokines and chemokines to exacerbate inflammation (*Lackey and Olefsky, 2016*; *McNelis and Olefsky, 2014*). Given the role of macrophages in obesity, sex differences in macrophages are of particular interest and have been demonstrated before (*Gal-Oz et al., 2019*; *Varghese et al., 2022*). Visceral fat contains more infiltrating macrophages and higher expression of inflammatory cytokines than subcutaneous fat (*Weisberg et al., 2003*; *Strissel et al., 2007*), and male visceral adipose tissues accumulate more macrophages than females (*Chen et al., 2021a*; *Chen et al., 2021b*). The presence of sex-steroid hormones, specifically estrogen, was postulated to contribute to sex differences in obesity (*Camporez et al., 2019*; *Palmer and Clegg, 2015*; *Keselman et al., 2017*; *Grove et al., 2010*; *Stubbins et al., 2012*; *Sullivan et al., 2005*; *Heine et al., 2000*). Alternatively, we and others have demonstrated intrinsic sex-specific differences in macrophages, independent of sex-steroid hormones (*Chen et al., 2021a*; *Shenoda et al., 2021*; *Singer et al., 2015*). Male macrophages are more migratory and inflammatory, while protection in females is associated with higher production of anti-inflammatory cytokines, such as IL-10 (*Chen et al., 2021a*; *Lainez et al., 2018*). Macrophage function and activation in the adipose tissue are guided by their ontogeny, the cytokine environment, as well as myriad factors such as hypoxia, metabolites, and lipids (*Chakarov et al., 2022*). CD11c[+] M1-like macrophages are activated through innate TLR2/4 receptors and produce proinflammatory mediators (e.g. TNFα, IL-6, and CCL2) that drive metabolic changes. Adipose tissue macrophages or metabolically activated macrophages can be distinguished from other proinflammatory macrophages by several different cell surface markers, although they produce proinflammatory cytokines as well (*Kratz et al., 2014*). On the other hand, a T helper type 2 (Th2) cytokine environment within the adipose tissue promotes metabolic homeostasis and protective CD206[+] M2 macrophages that suppress inflammation. Immune drivers of the Th2 cytokine environment for M2 macrophage activation include IL-4-producing eosinophils and innate lymphoid cell (ILC)-2 (*Wu et al., 2011*; *Brestoff et al., 2015*). It is now recognized that macrophage activation is far more complex than the M1/M2 paradigm (*Chakarov et al., 2022*; *Hill et al., 2018*; *Jaitin et al., 2019*). However, the M1/M2 macrophage paradigm is a useful framework to begin to address pathways that can be targeted for obesity pathogenesis, and whether these are influenced by sex.

The focus of this study was to identify sex-specific immune effectors that regulate obesity pathogenesis, focusing on the M2 macrophage signature gene Resistin-like molecule α (RELMα). RELMα is a small, secreted cysteine-rich protein that is expressed by macrophages primarily in response to Th2 cytokines, but can also be induced by hypoxia (*Pine et al., 2018*; *Lv and Liu, 2021*). RELMα has pleiotropic functions ranging from inflammatory or immunoregulatory to microbicidal roles (*Li et al., 2021*; *Harris et al., 2019*; *Krljanac et al., 2019*). Within the myeloid population, RELMα is preferentially expressed in monocyte-derived macrophages, and is important for monocyte differentiation, infiltration into other tissues and survival (*Sanin et al., 2022*; *Bain et al., 2022*). In the adipose

tissue, RELMα is a defining marker for perivascular macrophages and is co-expressed with CD206 and Lyve1 (*Chakarov et al., 2022*; *Jaitin et al., 2019*). A beneficial function for RELMα in metabolic disorders has been proposed; CD301b+ phagocytes promoted glucose metabolism and net energy balance through secretion of RELMα, and RELMα overexpression promoted cholesterol homeostasis in hyperlipidemic low-density lipoprotein receptor-deficient mice (*Kumamoto et al., 2016*; *Lee et al., 2014*). Based on these previous findings, the goal of this study was to investigate how sex and RELMα regulate diet-induced obesity and inflammation by employing RELMα-deficient mice and utilizing flow cytometry and single-cell sequencing of visceral adipose stromal vascular fraction (SVF) to identify sex-specific and RELMα-dependent targets of obesity.

## Results

### RELMα protects female mice from HFD-induced obesity and inflammation

Studies investigating sex differences show that female mice are protected, or have delayed, diet-induced obesity, unless aged or challenged by ovariectomy (*Lainez et al., 2018*; *Salinero et al., 2018*). In support of the critical role of macrophage polarization in sex-specific differences, previous studies demonstrated that protection in females is associated with Th2 cytokine-induced M2 polarization, for example CD206 expression, while males exhibit increased CD11c-positive 'proinflammatory' M1-like macrophages in the adipose tissue (*Chen et al., 2021b*). The secreted protein RELMα is a signature protein expressed by M2 macrophages, with regulatory functions in downregulating inflammation and promoting tissue healing. A role for RELMα in promoting metabolic homeostasis has also been reported (*Kumamoto et al., 2016*; *Lee et al., 2014*). Based on these studies, we hypothesized that female-specific protection from HFD may be influenced by RELMα. To examine systemic and local factors that may provide protection to females, serum and visceral adipose tissue homogenate were obtained from male or female mice on a control-fed (Ctr) or HFD-fed for over 12 weeks. Under both Ctr and HFD conditions, female mice had significantly higher RELMα in the serum than males, and in adipose tissue under Ctr diet. Exposure to HFD diminished adipose RELMα levels in both sexes (*Figure 1A*).

To determine the role of RELMα in obesity, we placed RELMα knockout (KO) mice on Ctr and HFD, and compared their response to matched wild-type (WT) controls (*Figure 1—figure supplement 1A*). RELMα deficiency did not affect Ctr or HFD weight gain in males, however, RELMα deficiency in females led to significantly increased weight gain on HFD compared to WT on HFD (*Figure 1B*). Whole-body weight, and visceral and subcutaneous adipose weights were similarly increased in WT and KO males on HFD (*Figure 1C,D*). However, RELMα deficiency only affected HFD-fed females, with significantly increased body weight, and visceral and subcutaneous adipose tissue mass compared to HFD WT females. Chemokines that change with exposure to HFD were assessed in the adipose tissue of these mice (*Figure 1E*). The monocyte chemoattractant CCL2 was significantly elevated in WT and KO HFD males, while it remained low in females regardless of diet, as demonstrated before (*Chen et al., 2021a*). On the other hand, females had higher levels of the anti-inflammatory IL-10 than males, as demonstrated before (*Chen et al., 2021a*; *Lainez et al., 2018*), as well as the regulatory T cell growth factor GM-CSF, and the Th2 cytokine IL-5. The higher level of IL-10 and GM-CSF in females was dependent on RELMα and was further decreased with exposure to HFD, while IL-5 was not detected with HFD. Inflammatory cytokines such as TNFα, IL-6, and IL-1β were also measured, but no RELMα-dependent differences with HFD exposure were observed. A larger animal cohort size may have provided more powered analysis and identified more adipose protein differences between groups especially with CCL2, which demonstrated the greatest variability. Nonetheless, the adipose protein profile indicates sex- and RELMα-dependent effects of diet-induced obesity, which correlates with increased proinflammatory CCL2, and decreased anti-inflammatory and Th2 cytokines, IL-10, GM-CSF, and IL-5, respectively.

Adipose tissue inflammation was next examined by flow cytometry of the SVF from the visceral adipose tissue (*Figure 1F*). RELMα deficiency and HFD resulted in significantly higher leukocyte frequency in the male SVF, demonstrating a role of RELMα in males (*Figure 1G*). WT females did not exhibit increased leukocyte frequency in adipose tissues with HFD, as shown previously (*Chen et al., 2021a*). Compared to HFD-fed WT females, RELMαKO females fed HFD had significantly more SVF

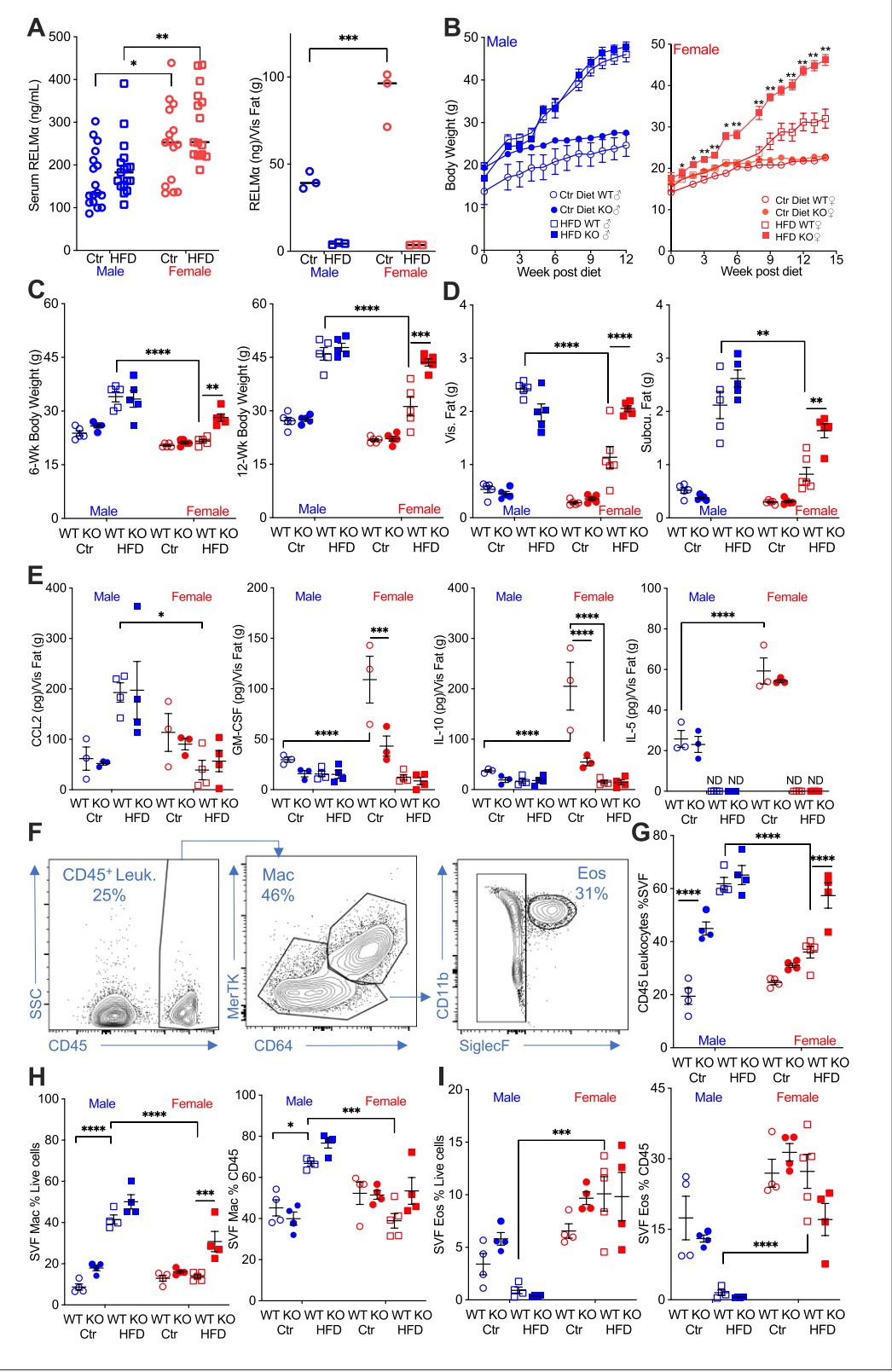

**Figure 1.** RELMα protects females from diet-induced obesity. (**A**) RELMα levels in serum and visceral adipose tissue from 18-week-old male (♂) and female (♀) C57BL/6 mice after exposure to control (Ctr) or high-fat diet (HFD) for 12 weeks. (**B**) Wild-type (WT) or RELMα knockout (KO) mice were weighed for 12–15 weeks of diet exposure. After 6- and 12-week diet exposure (**C**), whole body, visceral and subcutaneous fat pad weights were recorded (**D**).

*Figure 1 continued on next page*

*Figure 1 continued*

(**E**) CCL2, Granulocyte-macrophage colony-stimulating factor (GM-CSF), IL10, and IL5 levels in protein extracts from visceral fat pad after 12-week diet exposure. (**F**) Gating strategy for flow cytometric analysis of the visceral adipose stromal vascular fraction (SVF). Proportion in the SVF of CD45$^+$ leukocytes (**G**), CD45$^+$CD64$^+$Mertk$^+$ macrophages (**H**), and CD45$^+$SiglecF$^+$CD11b$^+$ eosinophils (**I**). Males (blue), females (red), WT (open symbols), RELMα KO (filled symbols), control diet (Ctr, circles), HFD (squares); data for (**B**) are presented as mean ± standard error of the mean (SEM), data for (**H**) are representative of one animal, all other data are presented as individual points for each animal, where lines represent group means ± SEM. Statistical significance between HFD WT females and HFD RELMα KO females was determined by two- or three-way analysis of variance (ANOVA) with Sidak's multiple comparisons tests (ND, not detected; *p < 0.05; **p < 0.01; ***p < 0.001; ****p < 0.0001 are indicated for functionally relevant comparisons). Data are representative of 3 experiments with 4–6 mice per group.

The online version of this article includes the following figure supplement(s) for figure 1:

**Figure supplement 1.** Experimental model figure and flow cytometry gating strategy.

leukocytes, specifically macrophages (*Figure 1H*). In contrast, the proportion of eosinophils was lower in HFD-fed males compared to females with the same diet (*Figure 1I*), suggesting a reduction in the protective type 2 immune response. This was consistent with the reduction in IL-5 in the adipose tissue following HFD (see *Figure 1E*). The contribution of diet, sex, and genotype to body weight and adipose tissue inflammation was assessed by three-way analysis of variance (ANOVA; *n* = 4–5 per group) (*Supplementary file 1*). Diet, followed by sex, then genotype, were all significant factors accounting for the variance in body weight, at both 6 and 12 weeks post diet. While diet was the greatest factor in adipose tissue inflammation, evaluated as SVF leukocyte frequency, RELMα deficiency was a greater factor accounting for variance than sex. For SVF macrophage frequency, diet then sex were the significant factors accounting for variance, while for eosinophil frequencies, sex differences were the main driving factor. Together, these data show that female-specific protection from diet-induced obesity is associated with elevated RELMα expression, and that RELMα deficiency selectively affects females, leading to increased weight gain, adipose tissue mass, and adipose tissue inflammation.

## RELMα deficiency results in dysregulated macrophage activation and impaired eosinophil homeostasis in the adipose tissue

We performed flow cytometry followed by t-distributed stochastic neighbor embedding (tSNE) analysis to evaluate immune cell heterogeneity and surface marker expression in the visceral adipose SVF (*Figure 2A*). tSNE analysis was performed based on gating strategies detailed in *Figure 1—figure supplement 1B*. Within the groups, eosinophils demonstrated the greatest changes; Ctr-fed male and female mice had high eosinophil numbers (see *Figure 1I*), and this eosinophil population disappeared in male mice upon HFD (*Figure 2B*, red outline). WT female mice retained their eosinophil subset even with HFD. In contrast, RELMα KO females had decreased eosinophil population following HFD. Within the eosinophil subset, heterogeneity is observed in females, with Ctr mice exhibiting a different cell distribution compared to the HFD mice. The macrophage population also exhibited changes; Ctr-fed female mice, had a smaller macrophage subset compared to males, regardless of genotype (*Figure 2B*, green outline). In both sexes, HFD led to an increase in this macrophage subset, and in their heterogeneity, especially in RELMα KO mice. Within the CD64$^+$MerTK$^+$ macrophage subset, expression of CD11c, a marker for proinflammatory M1-like macrophages, was evaluated. There was an increase in both number of CD11c+ macrophages and surface expression of CD11c on a per macrophage cell basis, in response to HFD in both males and females, however males had higher levels of CD11c than females under both diet conditions (*Figure 2C*). RELMα deficiency exacerbated the increase in CD11c, particularly in the HFD-fed females (*Figure 2D*). This may indicate that the increase in CD11c arises due to the abrogation of RELMα levels in the visceral fat after HFD (see *Figure 1A*). On the other hand, anti-inflammatory 'M2' macrophage marker, CD206, decreased with HFD, but was not dependent on the presence of RELMα (*Figure 2—figure supplement 1A*). CD301b, another M2 marker that is upregulated by IL-4, also decreased with HFD in both males and females (*Figure 2E*). Specifically in HFD females, CD301b was further decreased with a loss of RELMα. Although the number of eosinophils decreased in HFD-fed males of both genotypes, eosinophils in all groups maintained high expression of SiglecF, which was further increased with HFD specifically

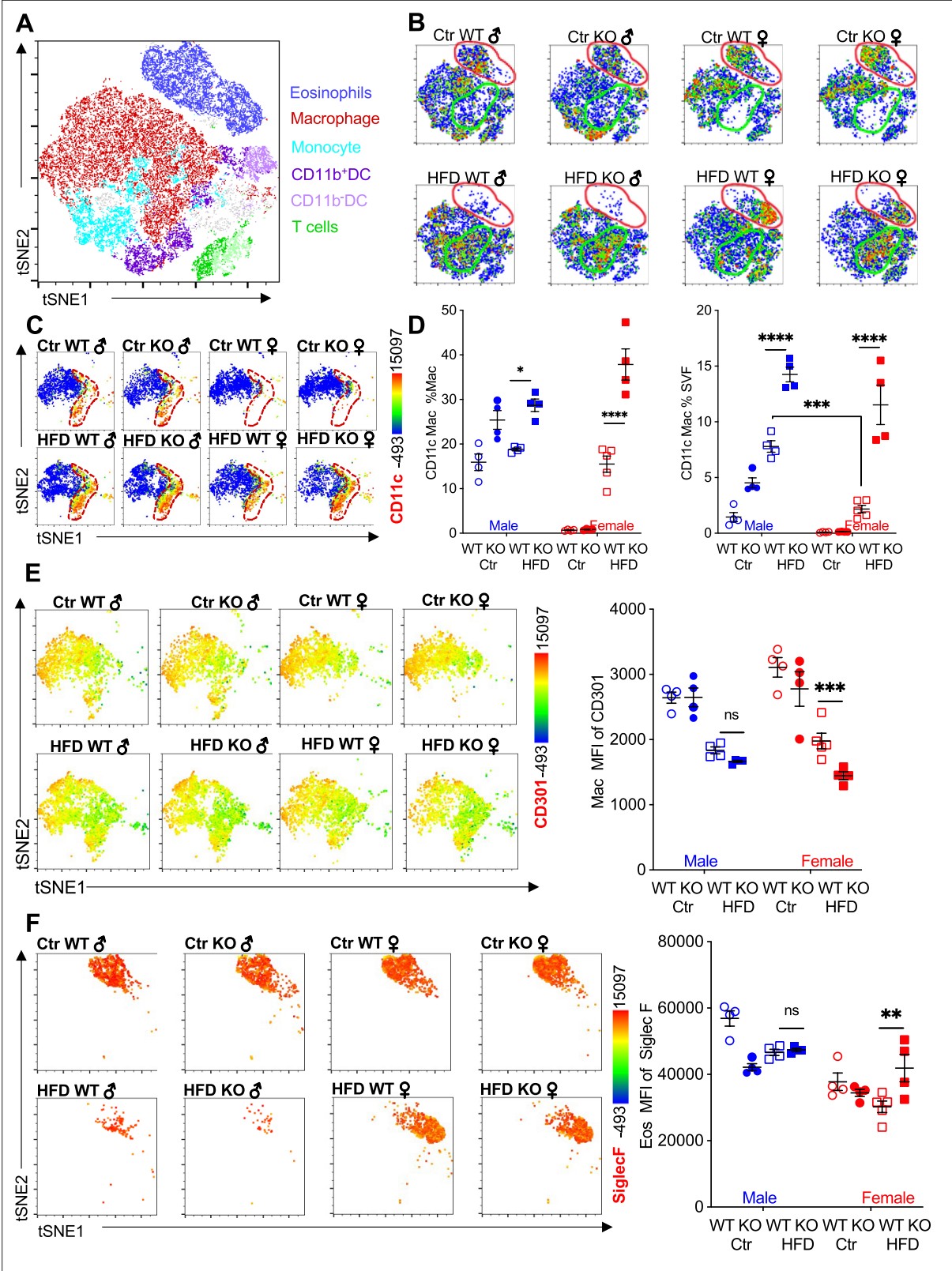

**Figure 2.** Adipose eosinophil and macrophage populations are influenced by sex, diet, and RELMα. (**A**) t-Stochastic neighbor embedding (tSNE) analysis to identify stromal vascular fraction (SVF) leukocyte populations. (**B**) tSNE analyses of SVF from the different groups (male ♂, female ♀, wild-type [WT], or RELMα knockout [KO]) after 12 weeks of diet exposure (Ctr or high-fat diet [HFD]) revealed changes in eosinophil RELMα-dependent and diet-induced changes in eosinophil (red outline) and macrophage (green outline) subsets. (**C, D**) CD11c surface expression in CD45+MerTK+CD64+

*Figure 2 continued on next page*

*Figure 2 continued*

macrophages was analyzed by tSNE, where dashed red outline shows CD11c^Hi cells, and quantified. (E) CD301b surface expression on SVF macrophages was examined by tSNE and quantified by mean fluorescent intensity (MFI). (F) Siglec-F surface expression on CD45+SiglecF+CD11b+ SVF eosinophils was examined by tSNE and quantified by mean fluorescent intensity (MFI). tSNE data are one representative animal per group. All other data are presented as individual points for each animal, where lines represent group means ± standard error of the mean (SEM). Statistical significance was determined by three-way analysis of variance (ANOVA) Sidak's with multiple comparisons test (ns, no significant; *p < 0.05; **p < 0.01; ***p < 0.001; ****p < 0.0001). Data are representative of 3 experiments with 4–6 mice per group.

The online version of this article includes the following figure supplement(s) for figure 2:

**Figure supplement 1.** Flow cytometric analysis of adipose immune cells.

in RELMα KO females but not WT females (*Figure 2F*). SiglecF is a paralogue of human Siglec-8, and in mice is expressed on eosinophils and alveolar macrophages. The function of SiglecF appears to be context dependent, with reported evidence of stimulatory and inhibitory roles on eosinophils (*Willebrand and Voehringer, 2016*; *Westermann et al., 2022*). One study showed that SiglecF stimulation induced apoptosis (*Zhang et al., 2007*). It is possible that the higher expression of SiglecF on the RELMα-deficient eosinophils from HFD KO female mice may contribute to their susceptibility to apoptosis, explaining their reduced frequency. We evaluated if eosinophil surface marker expression changed based on sex, diet, and genotype (*Figure 2—figure supplement 1B–D*). CXCR4 and MHCII expression was reduced following HFD in both WT and KO females but not males, which may account for the subset heterogeneity (see *Figure 2B*). Overall, these data identify that sex-specific and RELMα-dependent protection against diet-induced obesity is associated with changes in adipose macrophages and eosinophils.

## Protection against diet-induced obesity in females is mediated by RELMα and eosinophils

We evaluated whether associations existed between adipose immune cells and obesity by performing correlation analysis of body weight with adipose macrophage or eosinophil frequencies (*Figure 3A,B*). Across mice from all groups, there was a significant, positive correlation between macrophage frequency and body weight in the visceral SVF. In contrast, SVF eosinophil frequencies were negatively correlated with body weight. RELMα expression has been reported by many immune cell subsets, including macrophages, eosinophils, and B cells (*Pine et al., 2018*; *Chen et al., 2018*), although expression in the adipose tissue is less clear. Given that RELMα protein was present in the visceral adipose tissue, especially of females (see *Figure 1A*), flow cytometry analysis of intracellular RELMα in the SVF cells was performed in Ctr or HFD-fed female mice. SVF macrophages expressed RELMα, especially in the CD11c-negative subset, which was reduced with HFD (*Figure 3C,D*). RELMα+ SVF macrophage frequency was negatively correlated with body weight in females (*Figure 3E*), supporting the protective role of 'M2' macrophages in obesity. Focused comparisons between HFD vs. Ctr-fed female mouse groups showed that significant, negative correlation between RELMα+ macrophages and body weight occurred only in HFD-fed and not Ctr-fed mice (*Figure 3F,G*). These data raise the possibility that diet, rather than obesity per se, may be responsible for these significant correlations. Immunofluorescent staining of visceral adipose tissue sections was consistent with the flow cytometry analysis (*Figure 3H,I*); HFD-fed male mice had increased F4/80+ macrophage crown-like structures (green), while HFD-fed females had fewer F4/80+ macrophages but had more detectable SiglecF+ eosinophils (magenta). In contrast, eosinophils and RELMα were absent from HFD-fed RELMα KO females, which had increased F4/80+ cells compared to HFD-fed WT females. These data implicate RELMα-driven eosinophils as the underlying mechanism of female-specific protection from HFD-induced obesity and adipose tissue inflammation. This hypothesis was tested next by adoptive eosinophil transfer and recombinant RELMα treatment.

Following previously published methodologies for eosinophil adoptive transfer to protect against obesity (*Wu et al., 2011*; *Brestoff et al., 2015*), SiglecF+ eosinophils were column purified from WT female mice that were chronically infected with helminth *Heligmosomoides polygyrus*, to increase eosinophil frequency (*Figure 4A*). Phosphate-buffered saline (PBS) or eosinophils (Eos) were intraperitoneally transferred into HFD-fed WT or RELMα KO female mice every 14 days, and weight gain monitored for 7 weeks, followed by analysis of the peritoneal and visceral adipose tissue. As an alternative approach, RELMα KO female mice were treated with recombinant RELMα with the

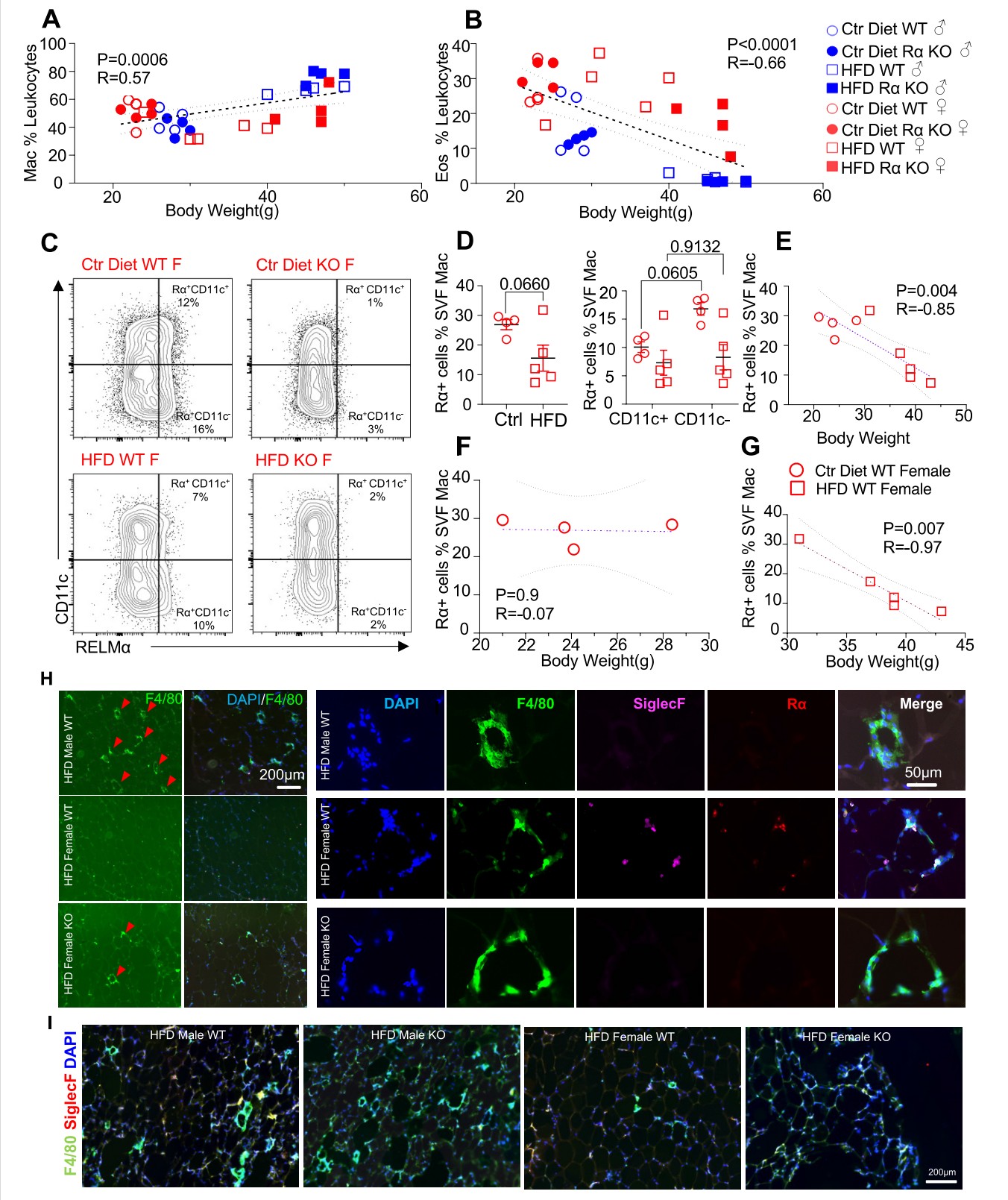

**Figure 3.** High-fat diet (HFD)-induced obesity is correlated with RELMα levels, eosinophils, and macrophages. Pearson correlation analysis of adipose stromal vascular fraction (SVF) macrophage (**A**) or eosinophil (**B**) frequency against body weight of mice from all groups. (**C**) Representative flow plots of RELMα intracellular staining against CD11c surface staining of SVF Mac from wild-type (WT) and knockout (KO) ♀ mice. (**D**) Frequency of RELMα⁺ SVF Mac in Ctr and HFD WT ♀ (left) or CD11c⁺ and CD11c⁻ Mac (right). (**E–G**) Pearson correlation analysis of RELMα⁺ cells against body weight of Ctr or HFD

*Figure 3 continued on next page*

*Figure 3 continued*

WT ♀ mice. (**H**) Immunofluorescent staining for F4/80 (green), SiglecF (magenta), RELMα (red), and DAPI, or 4',6-diamidino-2-phenylindole (blue) was counterstained on visceral fat tissue sections (bar, 200 μM; red arrows indicate F4/80⁺ cells). (**I**) IF staining was performed for F4/80 (green), SiglecF (red), and DAPI (blue) for all groups. Flow plots (**C**) and IF images (**H, I**) are one representative animal per group. All other data are presented as individual points for each animal, where lines represent group means ± standard error of the mean (SEM). Statistical significance was determined by unpaired *t*-test (**D**), or Pearson correlation analysis for other data and p values are provided. Data are representative of 2 experiments with 4–6 mice per group.

same timeline. As expected, PBS-treated RELMα KO females gained significantly more weight than PBS-treated WT mice, however, this was rescued by either eosinophil adoptive transfer or RELMα treatment, with the KO + Eos and KO + RELMα having equivalent body weight to WT + PBS and WT + Eos (*Figure 4A–C*). Flow cytometry analysis of the peritoneal cavity and visceral fat SVF confirmed reduced eosinophils in RELMα KO compared to WT mice, which was rescued by eosinophil transfer or recombinant RELMα treatment (*Figure 4D,E*). Evaluation of CD11c+ M1-like macrophages in the visceral fat confirmed that RELMα KO mice had more M1-like macrophages compared to WT mice, which were significantly decreased by either eosinophil transfer or RELMα treatment (*Figure 4F*). These data identify a RELMα–eosinophil–macrophage axis underlying female-specific protection from diet-induced obesity and inflammation; and strongly suggest that macrophage production of RELMα is necessary to promote adipose eosinophil homeostasis and inhibit M1-like macrophage activation, which is protective against HFD in females but not males.

## Single-cell RNA-sequencing of the adipose SVF uncovers sex- and RELMα-specific heterogeneity

To identify cell-specific gene expression changes underlying RELMα- and sex-dependent adipose effects, the 10× Genomics platform was used for single-cell RNA-sequencing (scRNA-seq) of the visceral adipose SVF from 6-week HFD-fed WT vs. RELMα KO, and males vs. females. At 6-week HFD, WT females were protected from weight gain, compared to the other groups (see *Figure 1B*), therefore this timepoint was chosen to define functionally relevant gene expression and pathway changes associated with weight gain. SVF single-cell suspensions from each mouse per group were labeled with cell multiplexing oligos (CMOs) to allow for pooling of biological replicates, prior to performing the single-cell 3' library generation and sequencing (*Figure 5A*). Principal component analysis (PCA) of all differentially expressed genes (DEGs) confirmed clustering of biological replicates by group (*Figure 5B*). A histogram of all DEG comparisons in all clusters between sex and genotype determined that WT male vs. WT female had the most DEG (*Figure 5—figure supplement 1A*). Because WT females are protected from diet-induced changes, we sought to analyze gene expression changes in all clusters in WT females compared to WT males, KO females, and KO males (*Figure 5—figure supplement 1B*). A heatmap of the top 30 genes showed that WT females have increased expression of serine/threonine kinase and proto-oncogene *Pim3*, and of anti-apoptotic gene *Bag3*, compared to the other three groups (*Figure 5—figure supplement 1C*). A Venn diagram of the top 100 genes in WT females compared to WT males, KO females, and KO males revealed that WT females uniquely upregulated 75 genes compared to the other three groups. The enriched pathways in the top 75 genes that were upregulated in WT females compared to the three other groups were sex specific (e.g. ovulation, sex differentiation, gonads, reproduction) (*Figure 5—figure supplement 2A,B*). Non-sex specific pathways that were enriched in protected WT females included vasculogenesis and response to lipids, providing molecular hints to genes that are associated with protection from obesity and inflammation, e.g. TGFβ (Tgfb1, Tgfbr2, Tgfbr3), Th2 cytokine signaling (Il4, Il4ra, Il13), and matrix remodeling (connexins, metallopeptidases).

The top DEGs and gene ontology (GO) pathways between males and females or WT and KO mice for all cells were examined (*Figure 5C–F*). Comparison of WT females vs. WT males showed that most highly DEGs are as expected *Xist* (the X inactivation gene in females) and *Ddx3y* (unique to males, expressed on the Y chromosome). Other most highly DEGs upregulated in females include *Il4ra*, suggesting M2 macrophage responsiveness, and genes involved in extracellular matrix, such as *Spon2*, which promotes macrophage phagocytic activity (*Figure 5C*). Males had higher levels of the sulfotransferase *Sult1e1*, and higher expression of inflammatory genes (e.g. *Lcn2*, lipocalin 2, and C7, complement 7). GO pathway analyses revealed that WT females upregulate genes in cellular responses to amyloid-beta. Compared to WT females, WT males had over-represented genes in terpenoid and

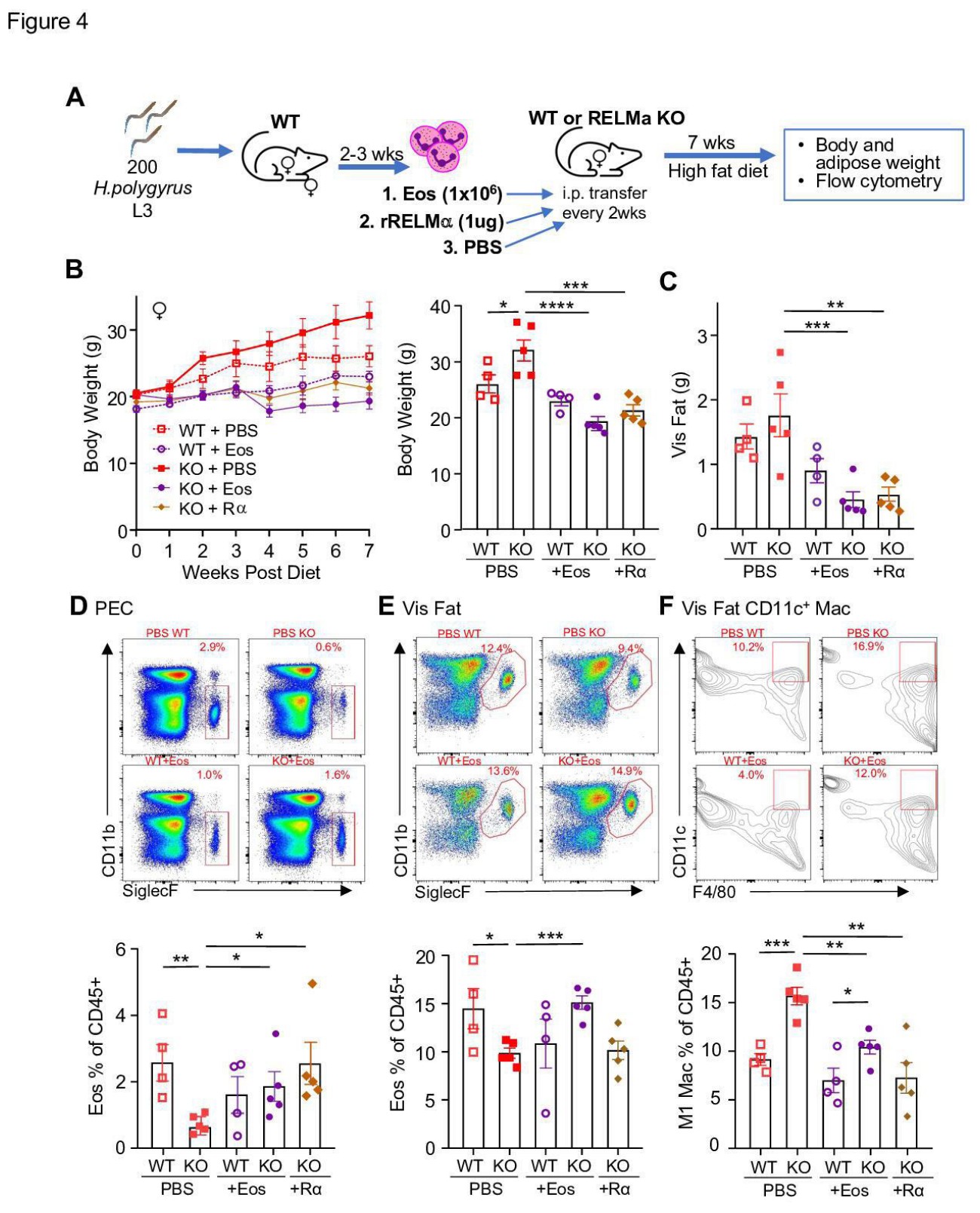

**Figure 4.** RELMα and eosinophils protect against diet-induced obesity. Wild-type (WT) or RELMα knockout (KO) female (♀) mice were exposed to high-fat diet (HFD) for 7 weeks, during which they were intraperitoneally injected every 2 weeks with phosphate-buffered saline (PBS), RELMα (2 μg) or SiglecF⁺ eosinophils (1 × 10⁶) recovered from helminth-infected WT ♀ mice (**A**). (**B**) Body weight was recorded every week. Mice were sacrificed at 7 weeks post diet, and body and visceral fat weight (**C**) were recorded. Flow cytometric analysis and quantification of eosinophils from the peritoneal

*Figure 4 continued on next page*

Figure 4 continued

exudate cells (PEC) (**D**), visceral fat stromal vascular fraction (SVF) (**E**), and quantification of the % of CD11c⁺ Macs in the visceral fat SVF (**F**). Data for (**B**) are presented as mean ± standard error of the mean (SEM), flow plots for (**D–F**) are representative of one animal per group, all other data are presented as individual points for each animal, where lines represent group means ± SEM. Statistical significance was determined by one-way analysis of variance (ANOVA) with Sidak's multiple comparisons test (ns, no significant; *$p < 0.05$; **$p < 0.01$; ***$p < 0.001$; ****$p < 0.0001$). Data are representative of 2 experiments with 4–6 mice per group.

isoprenoid biosynthetic pathway, which are involved in cholesterol synthesis. Comparison between KO males and KO females revealed shared sex-specific DEG compared to WT mice (***Figure 5D***; upregulated *Xist*, *IL4ra*, and downregulated *Sult1e1* in KO females). Unique female-specific genes that were also enriched were hemoglobin genes and oxygen-binding pathways. These were upregulated in KO female mice compared to KO males. In KO males, genes involved in extracellular matrix (Collagen 4 genes) and vascularization (*Ccn2*) were over-represented.

We then evaluated the RELMα-dependent genes that were associated with the loss of protection from diet-induced obesity in KO females (***Figure 5E***). Genes related to the negative regulation of amyloid proteins were the most enriched pathways in WT females compared to KO females, following a similar trend to the WT female vs. WT male comparison. The downregulation of this pathway in WT males and KO females suggest that sex and RELMα contribute to protection through this shared pathway. On the other hand, RELMα KO females compared to WT females upregulated hemoglobin genes and oxygen-binding genes. Of note, these RELMα-driven differences were unique to females since they were not identified in the comparison between WT vs. KO males. Instead, RELMα-regulated genes in males mapped to innate inflammatory response pathways (e.g. increased genes related to MHC Class 2, chemokine/chemokine receptor signaling in the KO males), while genes in cholesterol synthesis pathway were over-represented in WT males compared to KO males (***Figure 5F***). Of interest, long non-coding RNA *Gm47283/Gm21887*, located in the syntenic regions of both sex chromosomes (annotated as Gm47283 on Y chromosome, and Gm21887 on X chromosome), is the most upregulated RNA in both male and female RELMα KO mice compared to their WT counterparts (***Figure 5E,F***; Log2 fold change of 3.3 in KO males vs. Log2 fold change of 2.4 in KO females). Together, these data suggest that female-specific genes regulated by RELMα map to non-immune but hypoxic and iron stress-related pathways (hemoglobin, oxygen binding, and ferroptosis).

## Cell-specific gene expression changes in fibroblasts, ILC2, and myeloid subsets correlate with sex-specific and RELMα-dependent protection against diet-induced obesity

Cell-specific gene expression changes were evaluated. Based on expression of known marker genes, 12 clusters were identified, consisting of immune and non-immune cells, displayed as tSNE plots and histograms (***Figure 6A,B***). Eosinophils were not detected in any of the clusters. This was also shown in other SVF scRNA-seq studies, which concluded that eosinophils do not have sufficiently different transcriptomes from other leukocytes, or that there was a bias in the software, or technical difficulty such as low RNA content, or degranulation that leads to RNA degradation, which precluded eosinophil identification (***Weinstock et al., 2020***). At the same time as our ongoing analysis, the first publication of eosinophil single-cell RNA-seq was published, using a flow cytometry-based approach rather than 10×, that included RNAse inhibitor in the sorting buffer, and prior eosinophil enrichment (***Gurtner et al., 2023***). We employed targeted approaches to identify eosinophil clusters according to eosinophil markers (e.g. *Siglecf*, *Prg2*, *Ccr3*, *Il5r*), and relaxed the scRNA-seq cutoff analysis to include more cells and intronic content, but still could not detect eosinophils (***Figure 2—figure supplement 1G***). We concluded that eosinophils may be absent due to the enzyme digestion required for SVF isolation and processing for single-cell sequencing, which could lead to specific eosinophil population loss due to low RNA content, RNases or cell viability issues. Future experiments would be needed to optimize eosinophil single-cell sequencing, based on the recent publication of eosinophil single-cell sequencing.

The main population in the SVF, accounting for 50–75% of cells, were non-immune cells identified as *Pdgrfa*+ fibroblasts (green). They were significantly more abundant in WT females compared to the other groups ($p < 0.01$, S2C). We investigated if any subcluster in the fibroblast cell population expressed pre-adipocyte marker genes (Ppara, Pparg, Foxo1, Sirt1, Cebpa, and Cebpb) but no

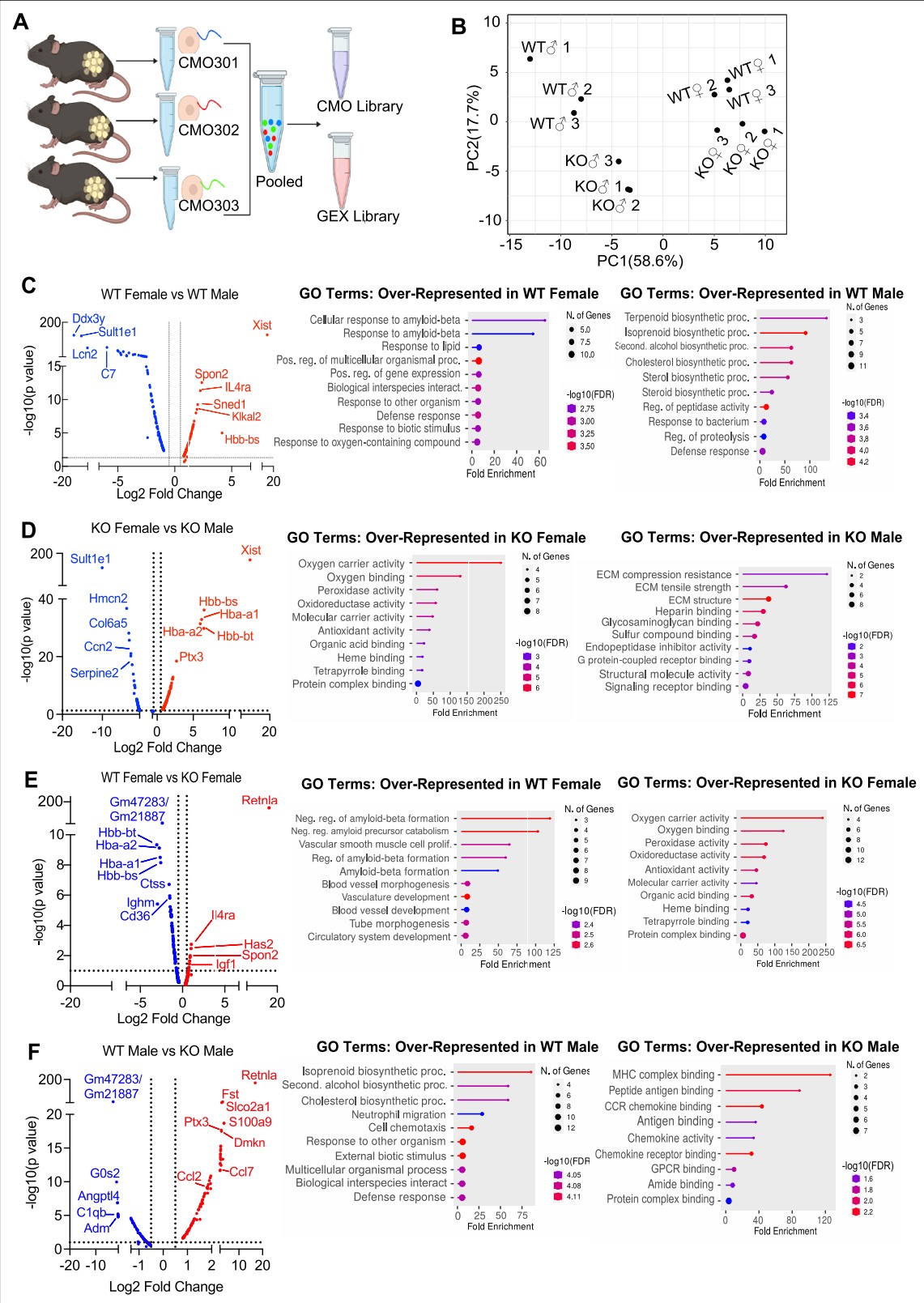

**Figure 5.** Single-cell RNA-sequencing (scRNA-seq) of adipose stromal vascular fraction reveals genes associated with protection from diet-induced obesity. Wild-type (WT) or RELMα knockout (KO) male (♂) or female (♀) mice were exposed to high-fat diet (HFD) for 6 weeks, following which cells from the adipose stromal vascular fraction were recovered for single-cell sequencing. (**A**) Schematic protocol of scRNA-seq cell multiplexing oligo (CMO) labeling and library preparation workflow. (**B**) Principal component analysis (PCA) assay of individual mice. Volcano plot comparing the top 100

*Figure 5 continued on next page*

Figure 5 continued

differentially expressed genes (DEGs) in all clusters between: WT females and WT males (**C**), KO females and KO males (**D**), WT females and KO females (**E**), and WT males and KO males (**F**). The most significant genes (−log10(p-value) >1, Log2 fold change >0.5) are indicated. Gene ontology (GO) terms indicating enriched pathways for the top 30 upregulated genes are plotted as histograms. Data are from 1 experiment with 3 mice per group.

The online version of this article includes the following figure supplement(s) for figure 5:

**Figure supplement 1.** Single-cell RNA-sequencing (scRNA-seq) of adipose stromal vascular fraction reveals differential gene expression between all four groups in all clusters.

**Figure supplement 2.** Differentially expressed genes (DEGs) in fibroblast and innate lymphoid cell (ILC)-2 cell populations from single-cell RNA-sequencing (scRNA-seq) analysis.

cluster-specific expression of these genes was observed (data not shown). Compared to WT males, SVF fibroblasts from WT females exhibited over-represented pathways involved in inhibition of cell proliferation and M2 macrophage responses (e.g. IL-4R, insulin growth-like factor, IGF-R) (*Figure 5— figure supplement 2C–L*). On the other hand, WT male fibroblasts upregulated genes involved in vasculogenesis and extracellular matrix deposition, such as collagen genes and *Ccn2*, which contributes to chondrocyte differentiation. Similar trends were observed between WT and KO females, indicating again that RELMα deficiency results in females that are more similar to males by adipose tissue gene expression. KO females had increased levels of fatty acid-binding proteins, *Fabp4* and connective tissue development, *Ccn3* and *Mmp3* (*Figure 5—figure supplement 2G–I*). In males, WT males SVF fibroblasts increased expression of prostaglandin transporter, *Slco2a1* and stromal chemokine *Cxcl12*, while KO males upregulated pathways involved in the inhibition of the antioxidative functions, resulting in the accumulation of reactive oxygen species and oxidative stress, such as *Txnip*. ILC2 are drivers of Th2 cytokine responses and are protective in obesity (*Brestoff et al., 2015*; *Ikutani and Nakae, 2022*; *Lee et al., 2015*). ILC2 were present at small frequencies in the SVF in all groups (*Figure 5—figure supplement 2M–P*). Gene expression analysis revealed that ILC2 from WT female mice expressed significantly higher Th2 cytokines (*Il13, Il5*) and *Csf2*, encoding for GM-CSF, which fits the increased adipose protein levels of IL-5 and GM-CSF in females (see *Figure 1*). Functional pathway analysis revealed that fatty acid metabolism genes were over-represented in WT females compared to males. When comparing WT and RELMα KO female mice, there was a reduction in *Csf2* in the KO females compared to WT females. These data indicate that ILC2 in females are functionally distinct from males, and may contribute to the protective Th2 cytokine environment and metabolic homeostasis in the adipose tissue.

Myeloid cells/macrophages were the main immune cell subset that changed in the SVF in response to sex and RELMα deficiency; macrophage proportions were lowest in WT females, but expanded in the other groups (*Figure 6A*, orange). GO analysis revealed that the IGF pathway, chemokine and cytokine activity pathways were over-represented in WT female myeloid cells, while innate immune activation (e.g. TLR-4, RAGE receptor) and extracellular matrix remodeling were higher in WT males (*Figure 6—figure supplement 1A*). These data match macrophage polarization signatures where protective M2 macrophages produce and are responsive to IGF, while M1 macrophages respond to danger signals (e.g. LPS, RAGE). RELMα-dependent changes were observed in both females and males (*Figure 6—figure supplement 1B*). Upregulated pathways in WT myeloid cells all involved innate chemokines and migration. Counterintuitively, downregulated pathways in WT compared to KO myeloid cells in females were associated with adipose tissue browning (e.g. brown fat cell differentiation, cold-induced thermogenesis), which are generally associated with protection from obesity. These data implicate RELMα in promoting innate immune cell migration and inhibiting adaptive thermogenesis. Overall, these scRNA-seq data identify sex-specific and RELMα-dependent changes in the adipose tissue that are associated with obesity-induced inflammation. Drivers of obesity included increased macrophages and innate immune activation. On the other hand, protection from obesity involved more fibroblasts, Th2 cytokine expression by ILC2, and chemokine expression by myeloid cells.

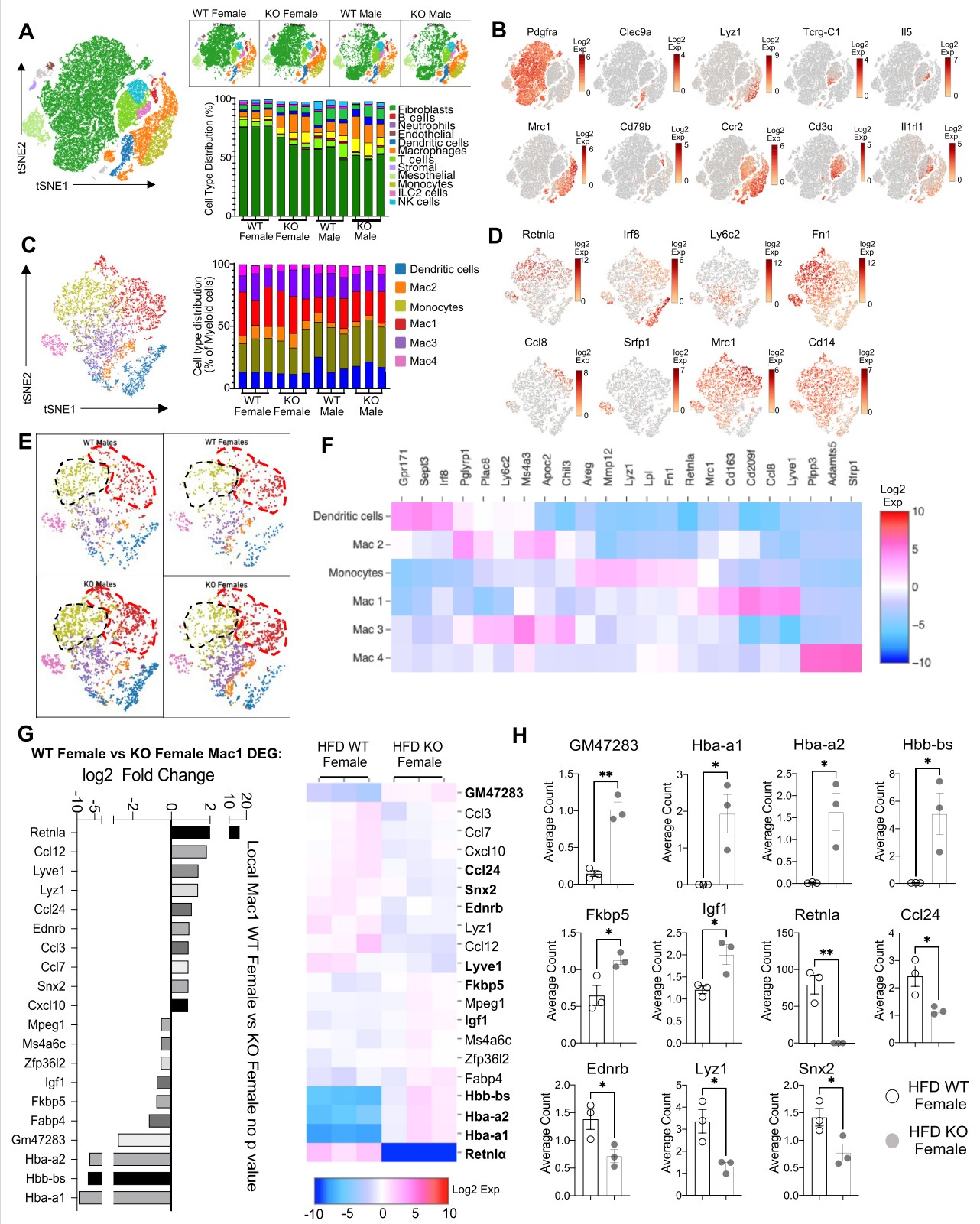

**Figure 6.** Sex-specific and RELMα-dependent gene expression changes in the stromal vascular fraction (SVF) myeloid subsets in response to high-fat diet (HFD). (**A**) t-Stochastic neighbor embedding (tSNE) plot showing cell populations from SVF from all four groups of mice fed HFD for 6 weeks, with a histogram plotting cell-type distribution per animal per group in all clusters. (**B**) Log2 fold change of candidate marker genes for each cell population across all clusters. (**C**) tSNE plot of re-clustered myeloid cell populations with a histogram plotting cell-type distribution per animal per group. (**D**) Log2

*Figure 6 continued*

fold change of candidate marker genes across myeloid cell populations. (**E**) tSNE plot highlighting population changes in Monocyte (green) and Mac1 (red) clusters between WT male, WT female, KO male, and KO female in myeloid cells. (**F**) Heatmap of the top differentially expressed gene (DEG) that defines each Mac subset. (**G**) WT female vs. KO female top DEG in Mac1 cluster. (**H**) Histograms of the average UMI count change of select candidate genes between WT female and KO female in Mac1 cluster. Data in (**H**) are presented as individual points for each animal, where lines represent group means ± standard error of the mean (SEM). Statistical significance was determined by unpaired *t*-test (*p < 0.05; **p < 0.01). Data are from 1 experiment with 3 mice per group.

The online version of this article includes the following figure supplement(s) for figure 6:

**Figure supplement 1.** Gene ontology (GO) pathway analysis for the top 30 hits in myeloid cell subsets.

**Figure supplement 2.** Monocyte and Mac1 differentially expressed gene (DEG).

## Monocyte-to-Mac1 macrophage transition and functional pathways are dysregulated in RELMα-deficient mice

Based on previous studies, macrophage subclusters were defined and enumerated according to the *Jaitin et al., 2019* study , and subset-specific gene expression was examined (*Figure 6C,D*). Comparison of the genes unique to each myeloid subset vs. the other subsets showed that monocytes were enriched for *Lyz1*, while Mac1 were lymphatic vessel-associated macrophages expressing *Lyve1* (*Figure 6F*). These two subsets in particular, were more abundant in WT males than WT females, and in KO males compared to KO females, but further increased in both sexes with a lack of RELMα (*Figure 6E*, black contour, monocytes; red contour, Mac1). Determined by scRNA-seq data from WT mice, RELMα (*Retnla*) exhibited higher expression in the Mono and Mac1 clusters (*Figure 6D, F*). We evaluated cell-intrinsic effects of RELMα on these subsets. GO analysis revealed strong enrichment for genes involved in leukocyte migration in both the Mono and Mac1 subsets from WT females, specifically eosinophil chemotaxis (*Figure 6—figure supplement 2A*). Evaluation of the top DEG indicated similar gene expression by sex rather than genotype (*Figure 6—figure supplement 2B*). Focused chemokine analyses identified sex and genotype-specific eosinophil-recruitment chemokines. In particular, the eosinophil-recruiting chemokine *Ccl24* was significantly reduced in Mono and Mac subsets from KO females compared to WT females. We examined the Mac1 subset in females, to identify RELMα-regulated genes within this population that typically expresses RELMα under normal conditions (*Figure 6G, H*). Hemoglobin genes were the most highly upregulated genes in KO female Mac1 cells compared WT females (5–10 Log2 fold change). The hypoxia-induced lncRNA *Gm47283/Gm21887* was also upregulated in KO female Mac1. The hemoglobin expression was not due to red blood cell (RBC) contamination for several reasons: first, mice were perfused and then RBC lysis was performed on the single-cell suspension; second, filtering was performed to remove doublets (see methods); third, single-cell analysis of the myeloid subsets for an RBC-specific gene (Gypa/CD235a) showed no expression, in contrast to the hemoglobin genes, which were expressed in the KO females (*Figure 7A,B*). To further validate that hemoglobin expression in macrophages was not tied to RBC contamination, IF staining for F4/80, hemoglobin, and RBC-specific marker Ter119, was performed on perfused adipose tissue sections from HFD-fed mice (*Figure 7C*). Hemoglobin protein was present (magenta), especially in RELMαKO females (white arrows), and co-localized with F4/80 (green). In contrast, there was little Ter119 staining, and minimal co-localization with hemoglobin, even at higher magnification (*Figure 7D*). Last, analysis of hemoglobin protein concentration in the adipose tissue lysates of WT and KO females was performed by ELISA, and determined significantly upregulated hemoglobin protein in KO females (*Figure 7D*). Together, these data indicate that RELMα deficiency induces hemoglobin genes in adipose tissue macrophages. Previous studies have shown that macrophages can upregulate hemoglobin genes during inflammation and hypoxia (*Liu et al., 1999*; *Saha et al., 2014*). Macrophage-specific upregulation of hemoglobin protein might indicate a response to hypoxia or oxidative stress in KO female Mac1 cells.

A trajectory analysis was performed to assess the relationships between the myeloid clusters, and whether they changed based on sex or genotype (*Figure 8A*). In WT females, monocytes were the point of origin, leading to the generation of Mac1 subsets. Mac2 and Mac3 were related but separate clusters. Dendritic cells and Mac4 were even more distinct from monocytes suggesting that they are resident and not monocyte derived. These trajectories were similar in WT males. However, in KO males and females, the clusters were no longer distinct, and the Mac1 cluster was able to become

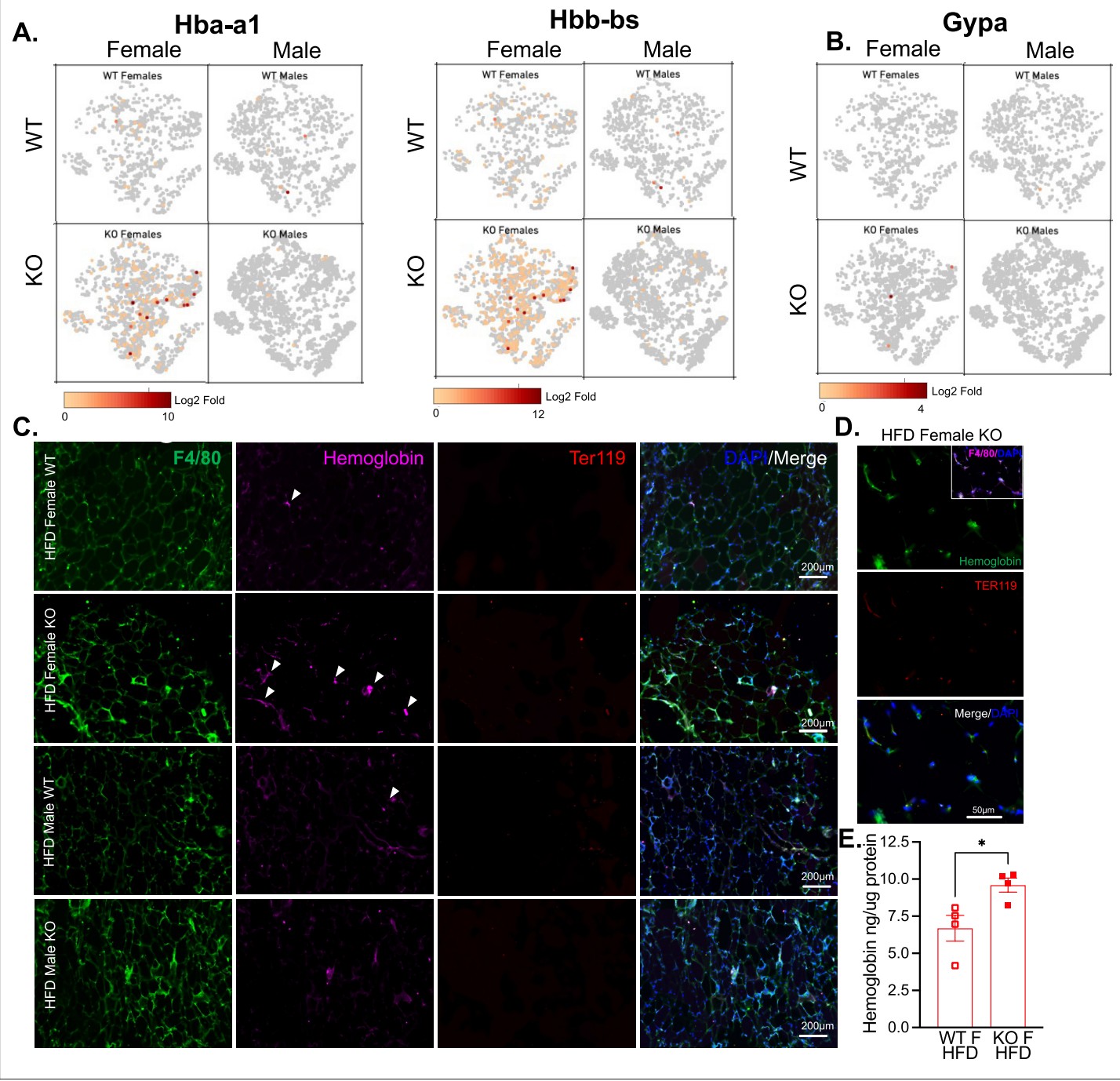

**Figure 7.** Hemoglobin expression in RELMα knockout (KO) myeloid cells. t-Stochastic neighbor embedding (tSNE) feature plots of Hba-a1 and Hbb-bs (**A**) and GypA (**B**) Log2 fold expression in myeloid clusters in WT female, KO female, WT male, and KO male. (**C**) Immunofluorescent staining for F4/80 (green), Hemoglobin (magenta), Ter119 (red), and DAPI (blue) was performed on visceral fat tissue sections (bar, 200 µM; arrows indicate Hemoglobin+ cells). (**D**) High magnification of high-fat diet (HFD) KO female. F4/80 (magenta, inset), Hemoglobin (green), Ter119 (red), and DAPI (blue) (bar, 50 µM). (**E**) Visceral adipose tissue homogenate hemoglobin ELISA in HFD WT or KO females. Data in (**E**) are presented as individual points for each animal, where lines represent group means ± standard error of the mean (SEM). Statistical significance was determined by unpaired *t*-test (*p < 0.05). Data are from 1 experiment with 3–4 mice per group.

Mac2/3 clusters, suggesting that loss of RELMα leads to dysregulated differentiation of monocytes to Mac1 or Mac2/3 subsets. We evaluated Mono to Mac1 transition in WT females and observed enriched pathways in IL-4 responsiveness and chemotaxis (*Figure 8B*). In contrast, Mono to Mac1 transition in RELMα KO females involved proton transport and ATP synthesis pathways, suggesting

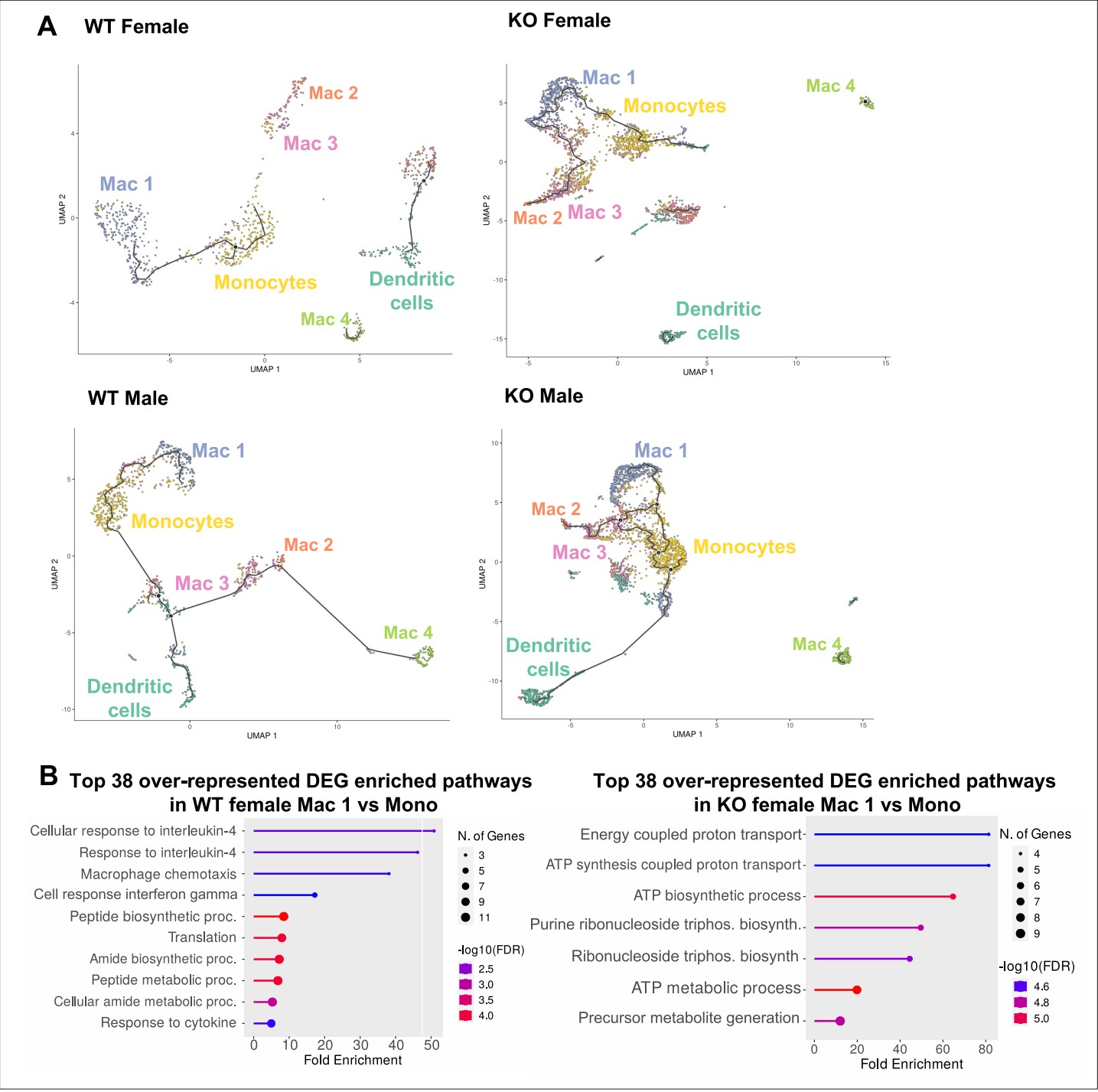

**Figure 8.** Trajectory analysis reveals dysfunctional myeloid differentiation in RELMα knockout (KO) females. (**A**) UMAP plots of trajectory analysis with monocytes set as the root were made for the myeloid subsets within each group (WT female, WT male, KO female, and KO male fed high-fat diet [HFD] for 6 weeks). (**B**) Histogram of top 38 differentially expressed genes (DEGs) with enriched gene ontology (GO) terms that were upregulated in WT female Mac 1 vs. Monocyte population and KO female Mac 1 vs. Monocyte population. Data are from 1 experiment with 3 mice per group (Supplementary material).

dysregulated differentiation leading to metabolically active, inflammatory Mac1 subsets. Together, these data implicate a critical function for RELMα in myeloid cell function and differentiation in the adipose tissue. First, we uncover a RELMα cell-intrinsic mechanism whereby RELMα-expressing Mono and Mac1 cells mediate leukocyte recruitment, and Mac1 preferentially recruits eosinophils. Second,

RELMα is necessary to drive functional Mac1 differentiation; in the absence of RELMα, Mac1 cells become metabolically active and increase their oxygen-binding capacity by upregulating hemoglobin genes. Given that Mac1 in the normal setting are defined as the protective, vascular-associated, and anti-inflammatory subset, the loss of function of this myeloid population may be the underlying mechanism for increased inflammation in RELMα KO mice.

## Discussion

The goal of this study was to investigate sex differences in obesity pathogenesis and elucidate immune mechanisms underlying female-specific protection from adipose inflammation, cardiovascular disease, and the metabolic syndrome. Herein, we uncover an eosinophil–macrophage axis in females that is driven by RELMα and protects from diet-induced obesity and inflammation. A role for RELMα in whole-body metabolism has been investigated (*Kumamoto et al., 2016*; *Lee et al., 2014*; *Munitz et al., 2009*), but none of these studies delineated sex-specific differences in chronic obese conditions. Here, by performing side-by-side comparisons between male and female RELMα KO or WT mice we identify sex- and RELMα-specific immune mechanisms of obesity pathogenesis. While C57BL/6J females are protected from obesity compared to males, we show that loss of RELMα abrogates this protection. RELMα deficiency also had significant effects in males, but to a lesser extent than females. For instance, RELMα KO males had increased proportions of leukocytes and CD11c$^+$ macrophages in the SVF to the same degree as exposure to HFD. Compared to WT females, RELMα KO females exhibited more diet-induced inflammatory changes than their male counterparts did. Under control and obese conditions, females had higher levels of RELMα than males, which likely explains why RELMα deficiency affected females more than males.

Several studies demonstrated the protective role of estrogen in obesity-mediated inflammation and in weight gain, as discussed above. Whether estrogen protection occurs via estrogen regulation of RELMα levels is a focus of our future studies. Alternatively, intrinsic sex differences in immune system have been demonstrated as well (*Chen et al., 2021a*; *Singer et al., 2015*) that are dependent on sex chromosome complement and/or *Xist* expression (*Syrett et al., 2018*; *Pyfrom et al., 2021*), and RELMα may be regulated by these as well. Additionally, aging-mediated increase in inflammation (including of adipose tissue, recently reviewed in *Zhang et al., 2023*), may also occur via changes in RELMα levels. Our studies used young but developmentally mature mice (4–6 weeks old when placed on diet, 18 weeks old at sacrifice), and future work on aged mice would be needed to investigate aging-mediated inflammation. Furthermore, there are sex differences in fat deposition, metabolic rates and oxidative phosphorylation (reviewed in *Mauvais-Jarvis, 2015*), and adipokine expression (*Chen et al., 2021a*; *Chen et al., 2021b*) which regulate cytokine and chemokines levels, and therefore may regulate levels of RELMα as well. These possibilities will be addressed in future studies.

A significant strength of this study was the use of single-cell sequencing to identify adipose tissue SVF heterogeneity and detect new targets and pathways to alleviate obesity. We first examined the top DEGs in protected WT females compared to the other groups and identify *Pim3* as protective. *Pim3* encodes a kinase that is a negative regulator of insulin secretion (*Vlacich et al., 2010*). *Pim3* is functionally responsive to forskolin, a cAMP activator that is also dietary supplement for weight loss and heart disease (*Godard et al., 2005*; *Mukaida et al., 2011*). Our data indicate that forskolin's effectiveness through Pim3 might be sex dependent. As the most significantly upregulated gene in the protected WT females, our data also point to cAMP activation as a promising target to alleviate diet-induced obesity. GO pathway analyses revealed that WT females upregulate genes in cellular responses to amyloid-beta. Amyloid-beta synthesis is elevated in obesity in humans (*Tharp et al., 2016*). In adipose tissue, it plays a role in lipolysis and secretion of adipokines (*Wan et al., 2015*). The increased cellular response to amyloid-beta specifically in females may explain female resistance to obesity-mediated changes. On the other hand, obese males had increased expression of *Sult1e1*, a sulfotransferase that leads to the inactivation of many hormones, including estrogen. *Sult1e1* is associated with increased BMI in humans (*Ihunnah et al., 2014*). Males also had higher expression of inflammatory genes (e.g. *Lcn2*, lipocalin 2, and C7, complement 7). WT males also upregulated genes in terpenoid and isoprenoid biosynthetic pathway, such as *Aldh1a3*, *Fdps*, and *Hmgcs1* that regulate cholesterol synthesis, triacylglycerol absorption and fat deposition. Their association with insulin resistance and metabolic syndrome may explain the male propensity to develop these diseases (*Castellano et al., 2020*). Examination of RELMα-dependent genes within the SVF led to the discovery of lncRNA

*Gm47283/Gm21887*, which is the most significantly induced RNA by RELMα deficiency in both males and females. This lncRNA is located in the syntenic region of sex chromosome; *Gm47283* on Y chromosome, while 100% identical *Gm55594* and *Gm21887*, are located on the X chromosome. Very little is known about *Gm47283*, apart from one recent paper indicating it is a biomarker for myocardial infarction that is induced in hypoxia and involved in prostaglandin 2 synthesis and ferroptosis (*Gao, 2022*). It is also called erythroid differentiation regulator 1 (Erdr1), which may correlate increase in this lncRNA in RELMα KO with hemoglobin gene induction (*Houh et al., 2016*). Our findings indicate that RELMα potently downregulates this lncRNA, and future research is warranted to investigate whether *Gm47283/Gm21887* is a downstream effector of RELMα.

Both control and HFD-fed females had a higher proportion of eosinophils in adipose tissues than males, and furthermore, males lost their eosinophil subset after exposure to HFD. Correlation analyses implicated that higher RELMα levels in females contributed to the higher proportion of eosinophils in female adipose tissues and female protection. This protection was lost in RELMα KO HFD-fed females, associated with the loss of eosinophils. Eosinophil transfer and RELMα treatment experiments confirmed this mechanistic link whereby RELMα recruits eosinophils with the overall outcome of reduced weight gain, decreased adipose tissue inflammation, and decreased CD11c$^+$ proinflammatory macrophages. These findings support the potential for RELMα treatment in males to protect from obesity-mediated inflammation by driving eosinophils. A critical function for eosinophils in establishing a Th2 cytokine environment in the adipose tissue has been reported (*Wu et al., 2011*; *Qiu et al., 2014*). Specifically, through use of eosinophil-deficient mice or transgenic mice that have increased eosinophils, these studies demonstrate that eosinophils produce IL-4 to promote M2 macrophages, which in turn mediate adipose tissue beiging and other protective pathways against obesity. In the context of helminths, studies also identified eosinophils as the underlying mechanism whereby helminth infection protects from obesity. Our studies uncover further complexity to eosinophil function by demonstrating that females have significantly increased adipose eosinophils, and that eosinophilia is critically dependent on RELMα. Since females do not gain weight compared to males, investigation of female-specific pathways in murine models of obesity is an understudied area. However, there is an urgent need to determine what mechanisms are protective in females and whether these change with age or menopause. These would allow the identification of new therapeutic targets and will also distinguish whether treatments may differ in their effectiveness according to sex. Our findings open a new area of investigation into RELM proteins, which are produced in humans, and whether they regulate eosinophils to protect from obesity. Another study investigated whether IL-5-induced eosinophils could protect obese male mice from metabolic impairments but reported no protective effects (*Bolus et al., 2018*). They concluded that physiological levels of eosinophils are not protective, in contrast to the previous studies that used transgenic mice to delete or artificially expand eosinophils. By additionally examining female mice and performing adoptive eosinophil transfer, our findings support a protective function for eosinophils even at physiologic levels, but also identify eosinophil heterogeneity. tSNE plot flow cytometric analysis of adipose SVF cells indicated that eosinophils were heterogeneous and different in females compared to males. These included changes in surface expression of CXCR4 and MHCII with HFD in females but not in males. In addition, single-cell sequencing analyses of the adipose SVF indicated striking sex- or RELMα-specific changes in multiple eosinophil chemoattractants such as IL-5, produced by ILC2, and myeloid cell-derived eotaxin-2 (CCL24) and CXCL10. Eotaxin-2 was produced by myeloid cells in the SVF female WT mice, but was significantly decreased with the loss of RELMα. Our data implicate female-specific and RELMα-dependent immune mechanisms in the adipose environment, whereby ILC2 and myeloid cells recruit eosinophils that function to downregulate obesity-induced inflammation.

Myeloid cells are critical for adipose tissue homeostasis, and monocyte recruitment and differentiation to proinflammatory macrophages are associated with obesity. Strikingly, RELMα deletion led to induction of hemoglobin genes in SVF female KO mice compared to WT females and KO male mice. This may have significant health implications. The importance of hemoglobin in erythrocytes is well accepted, but the presence of hemoglobin in non-erythroid cells is less well known with limited studies. Hemoglobin gene induction was first detected in RAW264 and isolated peritoneal macrophages (*Liu et al., 1999*). Alternatively, hemoglobin genes can be induced by iron-recycling macrophages, derived from Ly6c$^+$ monocytes during hemolysis, after erythrophagocytosis. Hemoglobin synthesis in cells other than erythroid lineage occurs in hypoxic conditions to increase oxygen

binding and compensate for low oxygen (*Grek et al., 2011*). Therefore, it is possible that the lack of RELMα in females leads to hypoxia in adipose tissues. Alternatively, hemoglobin may be induced in KO females in response to macrophage activation and nitric oxide (NO) production, since hemoglobin can bind NO in addition to oxygen (*Gell, 2018*), which is produced by activated macrophages (*Orecchioni et al., 2019*). Induction of hemoglobin genes may lead to dysregulation in iron handling and anemia, which have been associated with obesity. While obesity-increased incidence of anemia is not conclusive, iron deficiency is correlated with obesity (*Cepeda-Lopez and Baye, 2020*). Macrophages normally recycle iron, but lack of RELMα in obese females may have disrupted this ability. Increase in hemoglobin gene expression may lead to iron sequestration and would explain iron deficiency that is observed in obesity especially in women. Hemoglobin components include heme and iron, which can be cytotoxic. Overexpression of hemoglobin genes in the RELMα myeloid cells may not only act as a sink to deplete iron, oxygen, and heme with consequences for the SVF environment, but could also constitute cytotoxic stress for the myeloid cells themselves, in a positive feedback cycle spurring further adipose dysfunction. To our knowledge, this is the first evidence of a hemoglobin pathway in myeloid cells during metabolic dysfunction and may point to new therapeutic targets and biomarkers for adipose tissue inflammation and obesity pathogenesis.

RELMα function in peritoneal macrophages was previously demonstrated to be sexually dimorphic, where peritoneal macrophage replenishment from the bone marrow is lower in females, and macrophage differentiation in females, but not males, is RELMα dependent (*Bain et al., 2022*; *Bain et al., 2020*). Our data further reveal that RELMα expression is sex dependent and has critical functions in the adipose tissue through macrophage and eosinophil-driven mechanisms. We also demonstrate RELMα-specific effects on monocyte-to-macrophage transition in the adipose tissue that occur in both males and females. Whether these effects may be influenced by sex-specific differences in myeloid cell ontogeny from the bone marrow is unclear and an important avenue for future research. The importance of monocyte expression of RELMα for survival and differentiation has recently been reported (*Sanin et al., 2022*). Trajectory analysis of the myeloid subsets revealed that WT animals of both sexes followed expected trajectories of monocyte differentiation to either Mac1 or to Mac2/3 clusters. Mac2/3 clusters express markers of proinflammatory macrophages, such as Ly6c, while Mac1 expresses markers of anti-inflammatory macrophages, such as *Mrc1* (CD206). The lack of RELMα in KO animals of both sexes led to dysregulated monocyte differentiation, where the 'protective' Mac1 cluster could become Mac2/3 cells. This trajectory change implies that lack of RELMα disrupts myeloid differentiation leading to a more proinflammatory profile. Genes enriched in monocyte-to-Mac1 transition in WT vs. RELMα KO female mice were examined to determine cell-intrinsic functions for RELMα. These analyses revealed that RELMα expression is critical for monocyte differentiation into IL-4 responsive macrophages, but in its absence, monocytes begin to increase expression of genes associated with high metabolic activity, which could result in oxidative stress.

In conclusion, these studies demonstrate a previously unrecognized role for RELMα in modulating metabolic and inflammatory responses during diet-induced obesity that is sex dependent. Results from these studies highlight a critical RELMα–eosinophil–macrophage axis that functions in females to protect from diet-induced obesity and inflammation. Promoting these pathways could provide novel therapies for obesity pathology.

## Materials and methods

**Key resources table**

| Reagent type (species) or resource | Designation | Source or reference | Identifiers | Additional information |
|---|---|---|---|---|
| Strain, strain background (*Mus musculus*) | RELMα knockout | PMID:34349768 | | |
| Strain, strain background (*Heligmosomoides polygyrus*) | *H. polygyrus* | PMID:36569914 | | |
| Antibody | anti-Hemoglobin alpha (Rabbit monoclonal) | Invitrogen | Cat. # MA5-32328 | IF (1:100) |
| Antibody | anti-F4/80 (Rat monoclonal) | Invitrogen | Cat. # MA5-16624 | IF (1:100) |

*Continued on next page*

*Continued*

| Reagent type (species) or resource | Designation | Source or reference | Identifiers | Additional information |
| --- | --- | --- | --- | --- |
| Antibody | anti-TER-119- APC (Rat monoclonal) | eBioscience | Cat. # 17-5921-81 | IF (1:100) |
| Antibody | anti-RELM alpha-APC (Rat monoclonal) | Invitrogen | Cat. # 17-5441-82 | IF (1:100) |
| Antibody | anti-mouse CD170 (Siglec-F)-PE (Rat monoclonal) | BioLegend | Cat. # S17007L | IF (1:100) |
| Antibody | Anti-rabbit IgG cross-absorbed secondary-TRITC (Chicken polyclonal) | Invitrogen | Cat. # A15998 | IF (1:250) |
| Antibody | anti- F4/80-Alexa Fluor 488 (Rat monoclonal) | eBioscience | Cat. # 53-4801-82 | IF (1:100) |
| Antibody | anti-Rat IgG cross-absorbed secondary- Alexa Fluor 488 (Goat polyclonal) | Invitrogen | Cat. # A11006 | IF (1:250) |
| Antibody | anti-mouse CD16/CD32 (Rat monoclonal) | BD Biosciences | Cat. # 553141 | Flow (1:100) |
| Antibody | anti-mouse MERTK-FITC (Rat monoclonal) | BioLegend | Cat. # 151504 | Flow (1:200) |
| Antibody | anti-mouse CD25-PerCP (Rat monoclonal) | BioLegend | Cat. # 102028 | Flow (1:200) |
| Antibody | anti-mouse CD301- PerCP/ Cyanine5.5 (Rat monoclonal) | BioLegend | Cat. # 145710 | Flow (1:200) |
| Antibody | anti-mouse CD36-APC (Armenian Hamster monoclonal) | BioLegend | Cat. # 102612 | Flow (1:200) |
| Antibody | anti-mouse I-A/I-E- Alexa Fluor 700 (Rat monoclonal) | BioLegend | Cat. # 107622 | Flow (1:200) |
| Antibody | anti-mouse CD45-PerCP/Cyanine5.5 (Mouse recombinant) | BioLegend | Cat. # 157612 | Flow (1:200) |
| Antibody | anti-mouse CD184- Brilliant Violet 421 (Rat monoclonal) | BioLegend | Cat. # 146511 | Flow (1:200) |
| Antibody | anti-mouse/human CD11b- APC/ Cyanine7 (Rat monoclonal) | BioLegend | Cat. # 101226 | Flow (1:200) |
| Antibody | anti-mouse CD4- Brilliant Violet 711 (Rat monoclonal) | BioLegend | Cat. # 100557 | Flow (1:200) |
| Antibody | anti-mouse F4/80- Brilliant Violet 650 (Rat monoclonal) | BioLegend | Cat. # 123149 | Flow (1:200) |
| Antibody | anti-mouse CD206- Brilliant Violet 785 (Rat monoclonal) | BioLegend | Cat. # 141729 | Flow (1:200) |
| Antibody | anti-mouse CD170 (Siglec-F)- PE/ Dazzle 594 (Rat monoclonal) | BioLegend | Cat. # 155530 | Flow (1:200) |
| Antibody | anti-mouse CD11c-APC (Armenian Hamster monoclonal) | BioLegend | Cat. # 117310 | Flow (1:200) |
| Antibody | anti-mouse CD64-PE/Cyanine7 (Mouse monoclonal) | BioLegend | Cat. # 139314 | Flow (1:200) |
| Antibody | anti-mouse RELM alpha-PE (Rat monoclonal) | eBioscience | Cat. # 12-5441-82 | Flow (1:200) |
| Commercial assay or kit | Hemoglobin Elisa | abcam | Cat. # ab254517 | |
| Commercial assay or kit | Chromium Next GEM Single Cell 3' GEM Kit v3.1 | 10×Genomics | Cat. # 1000269 | |

*Continued on next page*

*Continued*

| Reagent type (species) or resource | Designation | Source or reference | Identifiers | Additional information |
|---|---|---|---|---|
| Commercial assay or kit | Chromium Next GEM Chip G Single Cell Kit | 10× Genomics | Cat. # 1000127 | |
| Commercial assay or kit | Dual Index Kit TT Set A, 96 rxns | 10× Genomics | Cat. # 1000215 | |
| Commercial assay or kit | 3′ CellPlex Kit Set A | 10× Genomics | Cat. # 1000261 | |
| Commercial assay or kit | Dual Index Kit NN Set A, 96 rxns | 10× Genomics | Cat. # 1000243 | |
| Software, algorithm | R | The R Foundation | RRID:SCR_001905 | V4.2.3 |
| Software, algorithm | Cell Ranger | 10× Genomics | | V7.0 |
| Software, algorithm | Cell Ranger multiplexing (multi) | 10× Genomics | | For use with Cell Ranger 6.0 and higher |
| Software, algorithm | Cell Ranger aggregation (aggr) | 10× Genomics | | Run cellranger multi prior |
| Software, algorithm | Seurat | Satija Lab PMID:31178118 | | V4.3 |
| Software, algorithm | Monocle3 | Cole-Trapnell Lab PMID:30787437 | | V3.1.2.9 |
| Software, algorithm | FlowJo | Treestar | | Version 10.8 |
| Other | High-fat diet (HFD) | Research Diets | Cat. # D12492 | Mouse food |
| Other | Control diet (Ctrl) | Research Diets | Cat. # D12450J | Mouse food |
| Other | Zombie Aqua Fixable Viability Kit | BioLegend | Cat. # 423102 | Viability dye (1:500) |
| Other | DAPI stain | Invitrogen | Cat. # D1306 | Nuclear stain (1 µg/ml) |

## Animals

All experiments were performed with approval from the University of California (Riverside, CA) Animal Care and Use Committee (A-20210017 and A-20210034), in compliance with the US Department of Health and Human Services Guide for the Care and Use of Laboratory Animals. RELMα KO mice were generated as previously described (*Li et al., 2021*). RELMα and their WT controls were maintained under a 12-hr light, 12-hr dark cycle and received food and water ad libitum. After weaning and a week acclimatization on normal chow, animals were randomly distributed in groups and placed on either an HFD (D12492, 60% kcal from fat; 5.21 kcal/g [lard 0.32 g/g diet, soybean oil 0.03 g/g], 20% kcal from carbohydrate, 20% kcal from protein; Research Diet, New Brunswick, NJ) or control diet with matching sucrose levels to HFD (Ctr, D12450J, 10% kcal from fat 3.82 kcal/g [lard 0.02 g/g diet, soybean oil 0.025 g/g], 70% kcal from carbohydrate, 20% kcal from protein; Research Diet, New Brunswick, NJ) for 6–12 weeks, as indicated for each experiment. For all tissue and cell recovery mice were sacrificed between 8 and 9 am.

## Eosinophil and RELMα treatment

For adoptive transfer, peritoneal exudate cavity cells were recovered from *H. polygyrus*-infected mice. Specifically, groups of 3–5 WT female BL/6 mice were infected by oral gavage of 200 *H. polygyrus* L3, which leads to adults in the intestine and eosinophilia by day 10 post-infection, and a chronic infection in BL/6J for at least 3 months (*Ariyaratne et al., 2022*). Infection was confirmed by egg count in feces. To ensure sufficient eosinophil recovery, 2–3 *H.polygyrus*-infected female mice were euthanized between days 14 and 20 post-infection for eosinophil recovery. Peritoneal exudate cavity eosinophils were column-purified with biotinylated anti-SiglecF (BioLegend), followed by anti-biotin MicroBeads then magnetic separation with MS columns according to the manufacturer's instructions (Miltenyi). 1 × 10$^6$ eosinophils were transferred to recipient mice by i.p. injection every 2 weeks. Eosinophil purity was confirmed by flow cytometry and by Diff-Quik stained-cytospins (*Figure 2—figure supplement 1E,F*). For RELMα treatment, recipient RELMα KO female mice were i.p. injected 2 µg RELMα every 14 days. Ctr mice were injected with PBS.

## Cytokine quantification

RELMα and IL5 were measured by sandwich ELISA. IFN-γ, CXCL1 (KC), TNF-α, CCL2 (MCP-1), IL-12p70, CCL5 (RANTES), IL-1β, CXCL10 (IP-10), GM-CSF, IL-10, IFN-β, IFN-α, and IL-6 were detected by the Mouse Anti-Virus Response Panel (13-plex) (Cat. # 740622 BioLegend, San Diego, CA) and analyzed on the NovoCyte Flow Cytometer (Agilent, Santa Clara, CA) and LEGENDplexTM software (BioLegend, San Diego, CA).

## Histological analyses and immunohistochemistry

At the conclusion of diet exposure, mice were anesthetized, perfused with 20 ml cold PBS, fat tissues were recovered and immersed in 4% paraformaldehyde (PFA) for 24 hr followed by 30% sucrose for another 24 hr. Fat tissues were embedded with O.C.T. (Sakura Finetek USA) and sectioned at 10 μm. For immunofluorescent staining, sections were incubated with APC-anti RELMα (DS8RELM, eBioscience, Santa Clara, CA), PE/Dazzle 594 anti-SiglecF (S17007L BioLegend, San Diego, CA) and Alexa Fluor 488 anti-mouse F4/80 (BM8 eBioscience, Santa Clara, CA) overnight at 4°C, then counterstained with DAPI (BioLegend, San Diego, CA). For Hemoglobin staining, sections were incubated with primary antibodies (rabbit anti-hemoglobin alpha (Invitrogen, catalog # MA5-32328)), F4/80 Monoclonal Antibody (Invitrogen, catalog # MA5-16624), TER-119 Monoclonal Antibody, APC (eBioscience, catalog # 17-5921-81) overnight at 4°C. Sections then washed with PBS-T three times, and then incubated fluorochrome-conjugated secondary antibodies (chicken anti-Rabbit IgG (H+L) Cross-Adsorbed Secondary Antibody, TRITC [Invitrogen, catalog # A15998]), Goat anti-Rat IgG (H+L) Cross-Adsorbed Secondary Antibody, Alexa Fluor 488 (Invitrogen, catalog # A11006) for 1 hr at RT. Sections were counterstained with DAPI (BioLegend, San Diego, CA). Slides were examined with the Keyence microscope (BZ-X800; lense:BZ-PF10P, Plan Fluorite 10X, WD 14.5 mm; BZ-PF40LP, Plan Fluorite 40X LD PH, WD 2.2–3.3 mm).

## Hemoglobin ELISA

The ELISA assay for Hemoglobin (Hba) was performed using a commercially available ELISA kit, Mouse Hemoglobin Elisa Kit (abcam, catalog # ab254517). Prior to measurement, total protein samples were diluted 1:1250. The absorbance data of the ELISA were acquired by a plate reader (BioTek Synergy HT).

## Flow cytometry

Tissues from each mouse were processed separately as part of a 3–4 mouse cohort per group, with each experiment repeated 2–3 times. In brief, mice were perfused with ice cold PBS, adipose tissue was collected from gonadal fat pads representing visceral fat depot, or from inguinal fat pads representing subcutaneous fat depot, rinsed in cold PBS, weighed, minced with razor blade and digested enzymatically with 3 mg/ml collagenase/dispase (Roche) at 37°C for 1 hr. Suspension was passed through 70 μm cell strainer, cells pelleted, and RBCs lysed using RBC lysis buffer (BioLegend, San Diego, CA). SVF cells were collected and counted, and 2 million cells labeled for flow cytometry analyses. Cells were Fc-blocked with anti-mouse CD16/CD32 (1:100, Cat# 553141, BD Biosciences, San Jose, CA) followed by surface marker staining with antibodies to MerTK (2B10C42, BioLegend), CD25 (PC61, BioLegend), CD301 (LOM-14, BioLegend), CD36 (HM36, BioLegend), MHCII (M5/114.15.2, BioLegend), CD45 (QA17A26, BioLegend), CXCR4 (L276F12, BioLegend), CD11b (M1/70, BioLegend), F4/80 (BM8, BioLegend), CD4 (RM4-5, BioLegend), CD206 (C068C2, BioLegend), SiglecF (S17007L, BioLegend), CD11c (N418, eBioscience), CD64 (X54-5/7.1, BioLegend), and RELMα (DS8RELM, eBioscience). Dead cells were labeled with Zombie Aqua Fixable Viability Kit (Cat# 423102 BioLegend, San Diego, CA). Gating strategy was followed: macrophage (CD45$^+$CD11b$^+$MerTK$^+$CD64$^+$), eosinophils (CD45$^+$CD11b$^+$SiglecF$^+$ MerTK$^-$CD64$^-$), monocyte (CD45$^+$CD11b$^+$MHCII$^+$CD11c$^-$MerTK$^-$CD64$^-$SiglecF$^-$), dendritic cells (CD45$^+$ MHCII$^+$CD11c$^+$MerTK$^-$CD64$^-$SiglecF$^-$), and T cells (CD45$^+$CD4$^+$CD11b$^-$MHCII$^-$CD11c$^-$MerTK$^-$CD64$^-$SiglecF$^-$). Cells were analyzed on the NovoCyte Flow Cytometer (Agilent, Santa Clara, CA) and FlowJo v10 software (Tree Star Inc, Ashland, OR). tSNE analyses were performed using FlowJo v10 (Tree Star Inc, Ashland, OR), following concatenation of samples (5000 cells per biological replicate) for each group, to generate plots consistent between groups. This was followed by analysis of the expression of desired markers in separated groups. The parameters used to run the tSNE analyses were FITC-MerTK, PerCP-CD25, Alexa Fluor 700-MHCII, Brilliant Violet

605-CD11b, Brilliant Violet 650-F4/80, Brilliant Violet 711-CD4, PE/Dazzle 594-SiglecF, PE Cy5-CD11c, and PE Cy7-CD64. Cells were gated according to *Figure 1—figure supplement 1B*, clustering was done according to these gates, and annotation was performed with the FlowJo software.

## SVF isolation from adipose tissue

Adipose tissue was dissected, rinsed in ice cold PBS, and minced with a razor blade. Fat was digested enzymatically with 3 mg/ml collagenase/dispase (Roche) at 37°C for 1 hr. Suspension was passed through a 70-μm strainer and centrifuged to pellet SVF. Cells were resuspended in PBS/0.04% bovine serum albumin, counted, viability determined to be >90% before proceeding to flow cytometry analysis or scRNA-seq.

## Single-cell RNA-seq

ScRNA-seq was performed following the Chromium Next GEM Single Cell 3′ v3.1 Dual Index with Feature Barcoding for Cell Multiplexing protocol. Each group consisted of three biological replicates (*Figure 5A*). SVF single-cell suspension from each mouse was labeled with the 10× Genomics CMOs following the manufacturer's protocol (10× Genomics, Demonstrated Protocol, CG000391). Cell suspensions from mice in the same group were pooled and processed for the Chromium Next GEM Single Cell 3′ v3.1 Dual Index (10× Genomics, Demonstrated Protocol, CG000388). For generation of single-cell gel beads in emulsion (GEM), 50,000 cells were loaded on the Chromium Chip and barcoded, in order to reach a targeted cell recovery of 30,000 cells per group. GEM reverse transcription was achieved in order to generate barcoded cDNA. GEMs were broken, and cDNA was cleaned up using DynaBeads MyOne Silane Beads (Thermo Fisher Scientific) and SPRIselect Reagent kit (Beckman Coulter). Full-length barcoded cDNA was amplified, cleaned up, and fragmented in order to generate Illumina-ready sequencing libraries. 3′ single-cell gene expression libraries were generated using a fixed proportion (25%) of the total cDNA per sample. Libraries were amplified by PCR, after which the library was split into two parts: one part for generating the 3′ gene expression library and the other for the multiplexing library. Libraries were indexed for multiplexing (Chromium Dual Index Kit TT Set A, PN-1000215, 10× Genomics), quantified by Qubit 3 fluorometer (Invitrogen), and quality assessed by 2100 BioAnalyzer (Agilent). Equivalent molar concentrations of libraries were pooled and sequenced using Novaseq 6000 (Illumina) using 10× Genomics recommended sequencing depth and run parameters (sequencing depth of 20,000 read pairs per cell, paired end sequencing) at the UC San Diego (UCSD) Institute for Genomic Medicine (IGM) Center.

## Data processing and analysis

Demultiplexed FASTQ files were provided and downloaded by a secure portal on the UCSD IGM core and were used for downstream processing. Raw scRNA-seq FASTQ files were aligned to the mouse mm10-2020-A genome with Cell Ranger v6.1.2 with default settings using STAR aligner in the Cell Ranger multiplex (multi) pipeline. The Cell Ranger multi pipeline specifically analyzes 3′ Cell Multiplexing data combined with 3′ Gene expression data. The reference genome was downloaded from the 10× Genomics website and built as per official release notes (here). Every group was analyzed using the Cell Ranger multi pipeline, which allows multiplexing libraries to be processed together with the paired gene expression (GEX) libraries for each group. CMO deconvolution was performed using the Cell Ranger multi pipeline. Briefly, a multi config CSV was created containing the library definitions and experimental design variables (deposited on the Github repository: https://github.com/rrugg002/Sexual-dimorphism-in-obesity-is-governed-by-RELM-regulation-of-adipose-macrophages-and-eosinophils copy archived at *Li, 2023*). These parameters contain sections that specify parameters relevant to analysis of the gene expression library including the 10× Genomics-compatible reference genome, a samples section that specifies sample information and CMO identity for cell multiplexing and a section that highlights the identity and location of the input FASTQ file for each sample as well as the chemistry of the assay. After generation of the multi config CSV, Cell Ranger multi was performed with the output folder containing the main pipeline outputs, such as the generalized multiplexing outputs and the demultiplexed outputs per sample. Dimensionality reduction analysis was performed as an automated secondary analysis as part of the Cell Ranger multi pipeline. Briefly, Cell Ranger performs PCA using gene expression features as PCA features, using a python implementation of the IRLBA algorithm. These data are then passed into the nonlinear dimensionality reduction method,

t-SNE analysis in order to visualize the data in a 2D space. Unbiased clustering of the data was then performed in order to group cells together that have similar expression profiles based on their principal components using a graph-based method. Cell Ranger produces a table indicating differentially expressed features in each cluster relative to other clusters, with the top hits being used to identify cell populations in *Figure 6*. Sequencing reads for all 12 samples were then integrated using the Cell Ranger aggregation (aggr) pipeline, which enables batch effect correction to be performed on the combined dataset. Cell Ranger aggr pipeline generates a normalized integrated count matrix was generated by dividing the UMI count for each gene by the total number of UMIs in each cell, followed by log-transformation. To filter out poor quality cells, cells with threshold UMI count of >20,000 and<500 and mitochondrial fraction of >10% were filtered from analysis using the Loupe Browser software v 6.2. re-clustering tool.

## Gene expression visualization and differential gene expression analysis

The 10× Genomics Loupe Browser software v 6.2 (10× Genomics, Pleasanton, CA) was used to project tSNEs of the cell-type clusters obtained after integration of all 12 samples (4 groups, WT female, WT male, KO female, KO male, *n* = 3 per group) using the Cell Ranger aggr pipeline.

Differential gene expression analysis for the myeloid cell (*Figure 6*), fibroblast, and ILC2 cell (*Figure 5—figure supplement 2*) subclustering analysis was calculated using the Loupe Browser's integrated locally distinguishing function, which determines the features that distinguish the selected groups from one another or by distinguishing cells from one selected cluster vs. cells from another selected cluster. The locally distinguishing function of Loupe Browser utilizes the negative binomial test based on the sSeq method (*Yu et al., 2013*), with Benjamini–Hochberg correction for multiple tests and calculates log-normalized average expression values across the two samples or cell populations being compared. When performing pseudobulk differential gene expression analysis between groups (*Figure 5*; WT female vs. male, KO female vs. male, KO female vs. WT female, KO male vs. WT male), the globally distinguishing function on Loupe Browser was used to find features between checked groups relative to all clusters in the dataset. Differential gene expression file outputs from Loupe Browser were downloaded and data were presented as volcano plots and average counts for select genes were plotted as histograms between both groups (GraphPad Prism). Heatmaps of differential gene expression data were plotted using Loupe Browser software and are generated using hierarchical clustering with Euclidian distance and average linkage.

GO enrichment analysis of the genes from DEG analysis was performed using the ShinyGo 0.76.3 platform (South Dakota State University [*Ge et al., 2020*]). ShinyGo 0.76.3 fold enrichment algorithm utilizes a hypergeometric distribution followed by false discovery rate (FDR) correction. Background gene sets are all protein-coding genes in the mouse genome. The top 30 DEGs that were up- and downregulated for each respective comparison were visualized using a dotplot chart plotting fold enrichment for each respective enriched GO term for Biological Processes. For each GO enrichment analysis, FDR of <0.05 was applied, with the pathway minimum set to 10.

## Trajectory analysis

For trajectory analysis, we utilized the Seurat package (Seurat_4.3.0) (*Stuart et al., 2019*) and Monocle3 (v 3_1.2.9) (*Cao et al., 2019*) to process and analyze the scRNA-seq data. The raw data were initially read into R using the readMM function, with the matrices for features, barcodes, and counts extracted from the input files. The Seurat object was created by applying the CreateSeuratObject function with a minimum of 3 cells per gene and 100 features per cell, which was then saved as an RDS file for subsequent analysis. Cell barcodes annotated from the 10× Genomics Loupe Browser were imported as separate CSV files. These annotations were subsequently integrated into the Seurat object's metadata. We then subset the data to focus on dendritic cells, macrophages, and monocytes for further trajectory analysis using Monocle3 (*Qiu et al., 2017*; *Haghverdi et al., 2018*). The expression matrix, cell metadata, and gene annotations were extracted from the Seurat object and used to create a new CellDataSet (CDS) object in Monocle3. The CDS was preprocessed with a dimensionality reduction set at 100 dimensions, followed by an alignment step to adjust for batch effects using the Sample.ID variable. Lastly, a trajectory graph was constructed by applying the learn_graph function, which infers the developmental trajectories of the cell populations (*Trapnell et al., 2014*).

## Code availability

Experimental protocols and the data analysis pipeline used in our work follow the 10× Genomics and Seurat official websites. The analysis steps, functions, and parameters used are described in detail in Methods. Deposition of code for cellranger multi, cellranger aggr, and trajectory analysis are deposited on the public github repository: https://github.com/rrugg002/Sexual-dimorphism-in-obesity-is-governed-by-RELM-regulation-of-adipose-macrophages-and-eosinophils copy archived at *Li, 2023*.

## Statistical analyses

Data are presented as mean ± standard error of the mean and statistical analysis was performed by GraphPad Prism 9. Statistical differences between control and RELMα KO mice ($p < 0.05$) were determined using *t*-test, or two- or three-way ANOVA with Sidak multiple comparisons test. *$p \leq 0.05$; **$p \leq 0.01$; ***$p \leq 0.001$; and ****$p \leq 0.0001$. 10× scRNA-seq experiment was performed once (3 mice per group). All other in vivo experiments were repeated 2–4 times with $n = 3$–5 per group (combined $n = 6$–20), based on sample size calculation by power analysis (Type I error <0.05 and Power ($1 - \beta$)).

## Acknowledgements

We thank Brandon Le for bioinformatics advice; Dr. Karine Le Roch for access to the 10× single-cell sequencing equipment; Constance Finney, Sang Woo, and Sang Yong Kim for provision of the Heligmosomoides polygyrus larvae. This research was supported by the National Institutes of Health (NIAID, R01AI153195 to MGN, NICHD R01HD091167 to DC). The data from this study were generated at the UC San Diego IGM Genomics Center utilizing an Illumina NovaSeq 6000 that was purchased with funding from a National Institutes of Health SIG grant (#S10 OD026929).

## Additional information

### Funding

| Funder | Grant reference number | Author |
| --- | --- | --- |
| National Institutes of Health | R01AI153195 | Meera G Nair |
| National Institutes of Health | R01HD091167 | Djurdjica Coss |
| National Institutes of Health | S10 OD026929 | Meera G Nair |

The funders had no role in study design, data collection, and interpretation, or the decision to submit the work for publication.

### Author contributions

Jiang Li, Rebecca E Ruggiero-Ruff, Conceptualization, Software, Formal analysis, Validation, Investigation, Visualization, Methodology, Writing - original draft, Writing - review and editing; Yuxin He, Formal analysis, Investigation, Methodology, Writing - review and editing; Xinru Qiu, Adam Godzik, Software, Formal analysis, Visualization, Methodology, Writing - review and editing; Nancy Lainez, Pedro Villa, Investigation, Methodology, Writing - review and editing; Djurdjica Coss, Conceptualization, Supervision, Funding acquisition, Investigation, Methodology, Writing - original draft, Project administration, Writing - review and editing; Meera G Nair, Conceptualization, Formal analysis, Supervision, Funding acquisition, Investigation, Methodology, Writing - original draft, Project administration

### Author ORCIDs

Rebecca E Ruggiero-Ruff http://orcid.org/0000-0001-9093-0233
Xinru Qiu http://orcid.org/0000-0003-1391-252X
Nancy Lainez http://orcid.org/0000-0002-3191-9023
Pedro Villa http://orcid.org/0000-0002-5245-4130
Adam Godzik http://orcid.org/0000-0002-2425-852X
Djurdjica Coss http://orcid.org/0000-0003-0692-1612

Meera G Nair ⓘ http://orcid.org/0000-0002-1807-5161

### Ethics

All experiments were performed with approval from the University of California (Riverside, CA) Animal Care and Use Committee (A-20210017 and A-20210034), in compliance with the US Department of Health and Human Services Guide for the Care and Use of Laboratory Animals.

### Decision letter and Author response

Decision letter https://doi.org/10.7554/eLife.86001.sa1
Author response https://doi.org/10.7554/eLife.86001.sa2

## Additional files

### Supplementary files

• Supplementary file 1. Three-way analysis of variance (ANOVA) of body weight, immune cells, macrophages, and eosinophils.

• Supplementary file 2. All enriched gene ontology (GO) terms for differentially expressed gene (DEG) analysis between WT female vs. WT male, KO female vs. KO male, KO female vs. WT female, and KO male vs. WT male in all cells.

• MDAR checklist

### Data availability

Sequencing data have been deposited in GEO under accession code GSE219119.

The following dataset was generated:

| Author(s) | Year | Dataset title | Dataset URL | Database and Identifier |
|---|---|---|---|---|
| Li J, Ruggiero-Ruff RE, Coss D, Nair MG | 2022 | RELMα provides sex-specific protection from obesity through macrophages and eosinophils | https://www.ncbi.nlm.nih.gov/geo/query/acc.cgi?acc=GSE219119 | NCBI Gene Expression Omnibus, GSE219119 |

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
