## [Editor Report]

In this study, Li and al describe valuable insights into mechanisms underlying sex-differences for diet-induced obesity in mice, demonstrating a role for macrophage-derived RELMa secretion in determining female-specific obesity protection. They provide evidence for the impact of RELMa signaling in eosinophil recruitment for diet-induced obesity protection in female mice. Single-cell RNA-seq analysis of the stromal vascular fraction of control and RELMa deficient animals were used to investigate molecular mechanisms underlying the protection. The conclusion of these findings provides evidence supporting dysregulation of cell differentiation pathways.

---

## [Decision Letter]

**Decision letter after peer review:**

Thank you for submitting your article "Sexual dimorphism in obesity is governed by RELMα regulation of adipose macrophages and eosinophils" for consideration by *eLife*. Your article has been reviewed by 3 peer reviewers, one of whom is a member of our Board of Reviewing Editors, and the evaluation has been overseen by Shlomo Melmed as the Senior Editor. The following individual involved in the review of your submission has agreed to reveal their identity: Sarah R. Ocanas (Reviewer #3).

Essential revisions:

After consulting together with the reviewer, we believe these are the essential revisions required to support the claims in the manuscript:

1. Increased clarity/details in the methodology, specifically in the processing and analysis of the scRNAseq dataset (including parameters, batch correction method, and immediate availability of code) [Rev 1 and 3].

2. Better discussion of why eosinophils could not be detected [Rev 1 and 3].

3. Trimming down conclusions from introductions and results, make it clear what is directly supported and what is conjecture [Rev 1, 2 and 3].

4. Check of mutilplexing/demultiplexing, since a top-regulated gene in females, Gm47283, is Y encoded [Rev 3]

*Reviewer #1 (Recommendations for the authors):*

1. The nature of the t-SNE analysis provided in Figure 2 is not well explained. What is the input data? How is the clustering performed? How are cell types annotated in the analysis? In general, neither the text nor the figure legend gives a good description of the content of Figure 2 in its current state.

2. There is a general lack of details on the analytical strategies and methods, especially computational methods, that is not acceptable for the evaluation of results and long-term reproducibility. The authors need to provide extensive additional information about used bioinformatic tools and/or analyses.

a. The authors give only very generic text about using "standard" data processing, which does not inform about specific algorithm choices for analysis. For instance, the authors mention using batch correction but do not specify the algorithm used (or its parameters). The pipeline for CMO deconvolution is also not described and needs to be described, especially for any future reanalysis of the data. In addition, the version of R, and all versions of R packages should be provided as well (e.g. monocle), as this can impact the final results. There is also no mention of which test was used to identify markers in Seurat, and how the tests were set up. Thus, it is crucial that the method section be revised to include all necessary information for readers to evaluate and reproduce the bioinformatics analyses of the scRNA-seq data.

b. Although a functional enrichment analysis is mentioned in the text [lines 230-232], there are no details about the algorithm used for enrichment beyond the name "shinyGO". Is this a hypergeometric enrichment or ranked list-type algorithm? What version of the GO database was used, and was it the 3 domains of GO? What background gene list was used to compute enrichment? All this information needs to be explicitly included. A supplementary table with all enriched terms would also be invaluable.

c. The code availability section uses language to say that the code is generic, following established tutorials. Since there are always choices made in the R implementation of a pipeline with a specific dataset (in terms of parameters), this is not acceptable for reproducibility or peer-review evaluation. In addition, it seems like different sources had to be used since batch correction is mentioned as well. For long-term reproducibility, it is crucial to either deposit all R scripts that were used to a public repository such as GitHub or provide it as a supplemental archive to accompany the manuscript. A reference to standard pipelines is not compatible with the methodological review.

d. The methods provide no description of the infection paradigm used to harvest eosinophils.

e. The authors do not provide any information about the time of day of euthanasia of animals, which impacts many phenotypes (including immune phenotypes) due to circadian rhythms.

3. Granulocytes, including eosinophils, are notoriously RNA-poor, and it has been established by 10xGenomics that special adjustments to the protocol (specifically, increased amplification of cDNA) are required for captured [https://kb.10xgenomics.com/hc/en-us/articles/360004024032-Can-I-process-neutrophils-or-other-granulocytes-using-10x-Single-Cell-applications-]. It would be important to discuss the RNA-poorness of these cells in the section discussing with eosinophils were not captured (lines 280-283), as this is more likely the reason for lack of capture over limited transcriptional identity or degranulation.

*Reviewer #2 (Recommendations for the authors):*

While we generally find the manuscript well written and well organized, there are some comments and concerns below we would like to share to help improve the manuscript:

Figure 1E: Small animal cohort size and large standard deviation in data. The figure would be strengthened by adding additional animals to improve the sample size.

Figure 1G: Differences in immune cell populations don't seem to correlate well with cytokine/chemokine expression (1E). Can classic cytokines known to increase during obesity like TNFa, IL-6, and IL-1b be looked at more carefully to better understand how the loss of RELMa influences their expression?

Figure 2A. Please state clearly in the body of the manuscript that this is flow cytometry data. This was not obvious from reading the text.

Figure 2E: Switch CD206 (main figure) and CD301 (supplementary).

Figure 2F: What is the significance of higher Siglec-F expression by eosinophils? please elaborate further on why this is important.

Line 206: It would be more proper to say M1-like macrophage, a classic M1 macrophage is defined by stimulation with LPS, and multiple papers over the years have shown CD11c+ macrophages in HFD models do not express classic M1 macrophage genes (PMID: 25242226).

Figure 4D-F does not seem to be representative flow plots based on the data graphed below. Recommend choosing a flow plot that represents the median of the graphed data.

Consider merging figures 5 and 6, as they are both limited in scope.

Figure 7a: Is the annotation able to differentiate fibroblast vs pre-adipocytes? Also in the Figure to the right of the TSNE plot, it is hard to determine the sex and genotype of these mice. Please draw groups on the x-axis like in 7C.

*Reviewer #3 (Recommendations for the authors):*

Below is a point-by-point critique to aid the authors in understanding specific points from the public review.

Title – We recommend including "in young mice" as part of the title.

Introduction – We believe that the introduction could benefit from editing down. Some of the details would be more appropriate for the discussion. Conclusions from the scRNA-Seq analyses will need to be updated after correcting the analyses.

Results – An experimental design figure prior to Figure 1 and Figure 4 would be helpful to the reader.

There are several instances in the text where the authors claim that there is a significant difference between the two groups, but the statistics for these comparisons are not shown in the figure.

– Line 97-98 – "Under both Ctr and HFD conditions, female mice had significantly higher RELMα in the serum and visceral adipose tissue than males (Figure 1A)." – It does not appear that there is a significant difference in RELMa within the female visceral fat under HFD.

– Figure 1B – Why was a t-test used for this figure when there were four groups to compare? Were any comparisons made between the Ctr groups? In the text (lines 100-101), it states "RELMα 101 deficiency did not affect Ctr or HFD weight gain in males…"; however these statistics are not described in the legend or text.

– Lines 107-108 – "The monocyte chemoattractant CCL2 was significantly elevated in HFD-males…" – Was this statistic assessed? It is not shown in Figure 1C.

– Lines 111-12 – "The higher level of IL-10 and GM-CSF in females were dependent on RELMα and were further decreased with exposure to HFD…" – In figure 1E, the effect of HFD was not shown to be statistically significant.

– Line 112 – "…while IL-5 was only decreased with HFD" – this statistic was not shown in Figure 1E.

– Figure 2B – Quantify and show statistics for the claim on lines 140-141 "Ctr-fed male and female mice had high eosinophil numbers, and this subset disappeared in male mice upon HFD".

– Figure 2E – It is difficult to determine which comparisons are being highlighted on the box plots. In the text (lines 154-156) the authors claim that this is an HFD effect – does this hold for both sexes?

It is unfortunate that eosinophils could not be identified in the single-cell analysis since this population of cells was shown to be important in rescuing the RELMα-deficiency in HFD-fed females. The authors should note in the discussion how future scRNA-Seq experiments could overcome this limitation (i.e., enriching immune cells prior to scRNA-Seq). Although another scRNA-Seq experiment including eosinophils would add to the report, we believe it is beyond the scope of this study.

– Lines 283-286 – "We employed targeted approaches to identify eosinophil clusters according to eosinophil markers (e.g. Siglecf, Prg2, Ccr3, Il5r), and relaxed the scRNA-Seq cutoff analysis to include more cells and intronic content, but still could not find eosinophils." – Please provide the feature plots showing expression across clusters for Siglecf, Prg2, Ccr3, Il5r as supplemental figures.

– Lines 286-288 – "We conclude that eosinophils may be absent due to the enzyme digestion required for SVF isolation and processing for single-cell sequencing, which could lead to specific eosinophil population loss due to RNases or cell viability issues." – Do you use the same single cell dissociation techniques for flow cytometry as were employed for the scRNA-seq? If so, what proportion of cells are eosinophils?

There is inconsistency in the presentation and interpretation of the correlations in Figure 3.

– The correlations in Figure 3A, 3B, and 3E appear to be driven entirely by diet effect. Is it possible that the HFD and not body weight are responsible for this effect?

– Figure 3C-G – Why are males absent from this analysis?

– Figure 3E – Why are all groups not included in this particular correlation, as in Figure 3A and Figure 3B? Is this a female-specific effect? I understand why the KO groups were not included, but the text does not describe the sex difference here.

– Lines 183-184 – "suggesting that the protective effect of RELMα may be impaired with body weight gain (Figure 3F-G)." – This result could also imply that HFD (and not body weight) is causing a decrease in RELMα. Do you think that this same effect would be seen in another obesogenic diet (i.e., a high carbohydrate diet)? We are not recommending adding these experiments to the current report, but authors may choose to comment on this in the discussion.

There are several issues with the scRNA-Seq analysis and interpretation. More details on the steps taken in the single-cell analyses should be included in the methods section.

– Deposition of code to a public repository (i.e., GitHub) would allow for rigor and reproducibility of the analyses.

With regards to the 'pseudobulk' analyses presented in Figures 5-6, several of the differentially expressed genes identified in Figure 6 are hemoglobin genes (i.e., Hba, Hbb genes). It is not uncommon to filter these genes out of single-cell analysis since their presence usually indicates red blood cell (RBC) contamination (PMID: 31942070, PMID: 35672358). We would recommend assessing RBC contamination as well as removing Figure 6 from the manuscript and focusing on cell-type-specific analyses.

– Figure 5D – It is unclear what comparisons are generating these lists of genes. Please provide a supplemental table with all differentially expressed genes from the analyses.

Within the text, there are several instances where the authors claim that a pathway is upregulated based on their Gene Ontology (GO) over-representation analysis (ORA). To come to this conclusion, the authors identify genes that are upregulated in one condition and then perform GO-ORA on these genes. However, the authors do not consider negative regulators, whose upregulation would actually decrease the pathway. Authors should either replace their GO-ORA analysis with one that considers the magnitude and direction of differentially expressed genes and provides an activation z-score (i.e., Ingenuity Pathway Analysis) or replace instances of 'upregulated' or 'downregulated' pathways with 'over-represented' pathways.

For Figure 7A, a representative tSNE plot for each group (WT Female, KO Female, WT Male, KO Male) should be shown to ensure there is proper integration of the clusters across groups. There are some instances where the scRNA-Seq data do not appear to be integrated properly (i.e., Supplemental Figure 2C). The authors should explore integration techniques (i.e., Seurat; PMID: 29608179) to correct for potential batch effects within the analysis.

– Lines 338-339 – "When analyzing scRNA-seq data from WT mice, RELMα (Retnla) is expressed in the Mono and Mac1 clusters (Figure 7D, F)." – It appears that RELMα is also expressed in the Mac4 cluster. Did you perform a plot to see if RELMα is present in any other clusters from Figure 7A?

LncRNA Gm47283 is identified as a gene that is differentially expressed by genotype in HFD females (Figure 7G); however, according to Ensembl this gene is encoded on the Y-chromosome (https://uswest.ensembl.org/Mus_musculus/Gene/Summary?g=ENSMUSG00000096768;r=Y:90796007-90827734). The authors should use the RELMα genotype and sex chromosomally-encoded genes to confirm that their multiplexing was appropriate.

For Figure 8, samples should be co-clustered and integrated across groups before performing trajectory analysis to allow for direct comparisons between groups.

– Line 289-291 – "The main population in the SVF, accounting for 50-75% of cells, were non-immune cells identified as Pdgrfa+ fibroblasts (green). These were significantly increased in WT females compared to the other groups (p<0.01, S2C)." – Were these data integrated using a feature like the "IntegrateData" command in Seurat? It does not appear that the cells cluster well when comparing males and females.

Scale bars on several heatmaps are not labeled with what they represent (Figures 5E, 7F, 7G, S2N, S2O, S4B, S4C).

Since the experiments presented in this report were from young mice using a single diet intervention, the authors should comment on how age and other obesogenic diets may impact the results found here. Also, the authors should expand their discussion as to what upstream regulators (i.e., hormones or genetics) may be driving the sex differences in RELMα expression in response to HFD.

---

## [Author Response]

Essential revisions:After consulting together with the reviewer, we believe these are the essential revisions required to support the claims in the manuscript:1. Increased clarity/details in the methodology, specifically in the processing and analysis of the scRNAseq dataset (including parameters, batch correction method, and immediate availability of code) [Rev 1 and 3].

We have significantly expanded the methodology, especially of the scRNAseq, and deposited the script and raw data in public repositories. This resubmission contains new Figure 7, and new supplementary figures and table (Supplementary File 2) with this methodology and validation.

2. Better discussion of why eosinophils could not be detected [Rev 1 and 3].

We have expanded this discussion and provided new citations.

3. Trimming down conclusions from introductions and results, make it clear what is directly supported and what is conjecture [Rev 1, 2 and 3].

We have significantly reduced the conclusions from these sections and removed any overinterpretation of the data.

4. Check of mutilplexing/demultiplexing, since a top-regulated gene in females, Gm47283, is Y encoded [Rev 3]

We have rigorously checked the single cell analysis, and multiplexing/demultiplexing, including discussion with 10x consultants and the director of the UCR Bioinformatics core. After this cross-checking, we can confirm that our results from the first submission are valid. All data are available for the readers, reviewers, and Editors, in public repositories. LncRNA Gm47283 is located in the syntenic region of the Y chromosome. It is also present in females, where due to the incomplete annotation, is named Gm21887, located in the syntenic region of the X chromosome. It also has 100% alignment with Gm55594 on X chromosome. To prevent confusion, we refer to it as Gm47283/Gm21887 in revision.

Reviewer #1 (Recommendations for the authors):1. The nature of the t-SNE analysis provided in Figure 2 is not well explained. What is the input data? How is the clustering performed? How are cell types annotated in the analysis? In general, neither the text nor the figure legend gives a good description of the content of Figure 2 in its current state.

We agree with the reviewer regarding the lack of detail for this tSNE analysis and provide clarity in two ways. First, we provide new panels (Figure 1—figure supplement 1B) detailing the gating strategy. Next, we provide this information in the methods: ‘tSNE analyses were performed using FlowJo v10 (Tree Star Inc.; Ashland, OR), following concatenation of samples (5000 cells per biological replicate) for each group, to generate plots consistent between groups. This was followed by analysis of the expression of desired markers in separated groups. The parameters used to run the tSNE analyses were FITC-MerTK, PerCP-CD25, Alexa Fluor 700-MHCII, Brilliant Violet 605-CD11b, Brilliant Violet 650-F4/80, Brilliant Violet 711-CD4, PE/Dazzle 594-SiglecF, PE Cy5-CD11c and PE Cy7-CD64. Cells were gated according to Figure 1—figure supplement 1, clustering was done according to these gates, and annotation was performed with the FlowJo software’

2. There is a general lack of details on the analytical strategies and methods, especially computational methods, that is not acceptable for the evaluation of results and long-term reproducibility. The authors need to provide extensive additional information about used bioinformatic tools and/or analyses.

We agree with the reviewer's suggestion to elaborate in more detail the methodology of our scRNA-seq experiments and data processing. We now provide extensive additions to our Materials and methods section detailing the process taken to achieve critical steps in our bioinformatic pipeline. We also divided the methods of our scRNA-seq approach into four sections: scRNA-seq, data processing, gene expression visualization and differential gene expression analysis and lastly, trajectory analysis. In addition, we have deposited all necessary codes for this analysis in our Github repository: https://github.com/rrugg002/Sexual-dimorphism-in-obesity-is-governed-by-RELM-regulation-of-adipose-macrophages-and-eosinophils. The addition of new information delineating the methodology of our work is highlighted in the manuscript.

a. The authors give only very generic text about using "standard" data processing, which does not inform about specific algorithm choices for analysis. For instance, the authors mention using batch correction but do not specify the algorithm used (or its parameters). The pipeline for CMO deconvolution is also not described and needs to be described, especially for any future reanalysis of the data. In addition, the version of R, and all versions of R packages should be provided as well (e.g. monocle), as this can impact the final results. There is also no mention of which test was used to identify markers in Seurat, and how the tests were set up. Thus, it is crucial that the method section be revised to include all necessary information for readers to evaluate and reproduce the bioinformatics analyses of the scRNA-seq data.

We have now provided a paragraph on the CMO deconvolution methods using the 10x Cell Ranger multiplex (multi) pipeline, a pipeline that is specific to CMO calling and deconvolution using 3’ CellPlex. Markers for identified clusters were identified using Cell Ranger, where each cluster contained a list of top differentially expressed genes. From that list we were able to identify cell type populations based on known markers in cell populations, examples of which are included in Figure 6B. As mentioned now in detail in our Materials and methods section, we performed principal component analysis, dimensionality analysis, CMO deconvolution, integration and differential gene expression using the Cell Ranger v.6.1.2 software. Information on unique cell features, barcodes and matrices were then exported and used to make a Seurat object for trajectory analysis using Monocle 3.

b. Although a functional enrichment analysis is mentioned in the text [lines 230-232], there are no details about the algorithm used for enrichment beyond the name "shinyGO". Is this a hypergeometric enrichment or ranked list-type algorithm? What version of the GO database was used, and was it the 3 domains of GO? What background gene list was used to compute enrichment? All this information needs to be explicitly included. A supplementary table with all enriched terms would also be invaluable.

In response to the reviewer’s suggestion to include more details on the algorithm used for gene ontology enrichment analysis, we added information on the R package that we used: We utilized ShinyGo 0.76, an R Package developed by Dr. Ge at South Dakota State University, based on several R packages for in-depth analysis of gene lists with options of several graphical visualization of enrichment and pathway information. Source code is reported in the publication, Steven Xijin Ge, Dongmin Jung, Runan Yao, ShinyGO: a graphical gene-set enrichment tool for animals and plants, Bioinformatics, Volume 36, Issue 8, April 2020, Pages 2628–2629, and is available on the Github repository (https://doi.org/10.1093/bioinformatics/btz931). This is a hypergeometric enrichment algorithm. We used the ShinyGo 0.76, which is based on Ensembl Release 104 with revision, archived on September 2, 2022. ShinyGo 0.76 uses the GO database version V2017.5, consisting of 11,943 GO terms for Biological Processes. Please also see Supplementary File 2, which we added as suggested by the reviewer.

c. The code availability section uses language to say that the code is generic, following established tutorials. Since there are always choices made in the R implementation of a pipeline with a specific dataset (in terms of parameters), this is not acceptable for reproducibility or peer-review evaluation. In addition, it seems like different sources had to be used since batch correction is mentioned as well. For long-term reproducibility, it is crucial to either deposit all R scripts that were used to a public repository such as GitHub or provide it as a supplemental archive to accompany the manuscript. A reference to standard pipelines is not compatible with the methodological review.

We have now deposited all R scripts and other necessary code to a public Github repository for transparency and reproducibility (https://github.com/rrugg002/Sexual-dimorphism-in-obesity-is-governed-by-RELM-regulation-of-adipose-macrophages-and-eosinophils ). We also deposited the R script for the trajectory analysis on GitHub repository. In addition, we also added the multi configuration CSV and code for the Cell Ranger multi pipeline as well as the code for the Cell Ranger aggregation (aggr) pipeline performed on Cell Ranger. Information on Cell Ranger and the pipelines used in this manuscript, aggregation (aggr) and multiplexing (multi) are highlighted in this technical support note by 10x Genomics: https://support.10xgenomics.com/single-cell-gene-expression/software/pipelines/latest/what-is-cell-ranger

Cell Ranger aggr pipeline:

https://support.10xgenomics.com/single-cell-gene-expression/software/pipelines/latest/using/multi

Cell Ranger aggr pipeline:

https://support.10xgenomics.com/single-cell-gene-expression/software/pipelines/latest/using/aggregate

d. The methods provide no description of the infection paradigm used to harvest eosinophils.

We have provided more information in the Methods section, provided a diagram in Figure 4 (new Figure 4A), and referred to our recent publication on H.polygyrus. Additional text in Methods: ‘For adoptive transfer, peritoneal exudate cavity cells were recovered from Heligmosomoides polygyrus-infected WT female BL/6 mice. Specifically, groups of 3-5 WT female BL/6 mice were infected by oral gavage with 200 H.polygyrus L3, which leads to adult worms in the intestine and eosinophilia by day 10 post-infection, and a chronic infection in BL/6J for at least at least 3 months (PMID: 36569914). Infection was confirmed by egg count in feces. To ensure sufficient eosinophil recovery, 2-3 H.polygyrus-infected female mice were euthanized between days 14 and 20 post-infection for eosinophil recovery. Peritoneal exudate cavity eosinophils were column-purified with biotinylated anti-SiglecF (BioLegend), followed by anti-biotin MicroBeads then magnetic separation with MS columns according to manufacturer’s instructions (Miltenyi). 1x10^6^ eosinophils were transferred to recipient mice by i.p. injection every two weeks.’

e. The authors do not provide any information about the time of day of euthanasia of animals, which impacts many phenotypes (including immune phenotypes) due to circadian rhythms.

We have clarified that euthanasia occurred between 8am and 9am in the morning for all experiments. In methods: ‘For all tissue and cell recovery mice were sacrificed between 8 and 9am.’

3. Granulocytes, including eosinophils, are notoriously RNA-poor, and it has been established by 10xGenomics that special adjustments to the protocol (specifically, increased amplification of cDNA) are required for captured [https://kb.10xgenomics.com/hc/en-us/articles/360004024032-Can-I-process-neutrophils-or-other-granulocytes-using-10x-Single-Cell-applications-]. It would be important to discuss the RNA-poorness of these cells in the section discussing with eosinophils were not captured (lines 280-283), as this is more likely the reason for lack of capture over limited transcriptional identity or degranulation.

We agree with the reviewer that granulocytes are RNA-poor. Our primary focus was to determine differences between all groups in ALL immune cells in the stromal vascular fraction that arise due to obesity. We were concerned to deviate from the standard protocol and increase amplification due to possible amplification bias, which may prevent comparison with previous studies. We had discussions with other researchers and 10X genomics concerning granulocyte populations, and opted to use this protocol to gain the most insight into all immune cells, especially myeloid lineage that is critical in obesity, even if it diminished granulocyte RNA. We included more discussion on this point, including citing a recent paper that is the first to our knowledge to report single cell sequencing in eosinophils: “At the same time as our ongoing analysis, the first publication of eosinophil single cell RNA-seq was published, using a flow cytometry based approach rather than 10x, including RNAse inhibitor in the sorting buffer, and performing prior eosinophil enrichment (PMID: 36509106). We employed targeted approaches and relaxed the scRNA-Seq cutoff analysis to include more cells and intronic content to attempt eosinophil identification using eosinophil markers (e.g. Siglecf, Prg2, Ccr3, Il5r), but still could not find eosinophils – see new panel Figure 2—figure supplement 1G. We conclude that eosinophils may be absent due to the enzymatic digestion required for SVF isolation and processing for single cell sequencing, which could lead to specific eosinophil population loss due to low RNA content, RNases or cell viability issues. Future experiments would be needed to optimize eosinophil single cell sequencing, based on the recent publication of eosinophil single cell sequencing”.

Reviewer #2 (Recommendations for the authors):While we generally find the manuscript well written and well organized, there are some comments and concerns below we would like to share to help improve the manuscript:Figure 1E: Small animal cohort size and large standard deviation in data. The figure would be strengthened by adding additional animals to improve the sample size.

We agree with the reviewer that increasing the number of samples would likely decrease standard deviation for CCL2, while the other cytokines do not exhibit variability. However, since CCL2 is not regulated by RELMa, and differences between males and females have already been demonstrated by our group (PMID:33268480), we do not think that it is justifiable to sacrifice additional animals for confirmatory studies, and it would not comply with 3 R’s requirement for animal research. We have added a statement acknowledging the caveat of this small animal cohort size.

Figure 1G: Differences in immune cell populations don't seem to correlate well with cytokine/chemokine expression (1E). Can classic cytokines known to increase during obesity like TNFa, IL-6, and IL-1b be looked at more carefully to better understand how the loss of RELMa influences their expression?

We performed a Biolegend Legendplex array to detect 12 cytokines/chemokines, which include TNFa, IL-6 and IL-1beta, but did not observe RELMalpha-specific differences in the HFD mice Therefore we chose not to include these for simplicity. We now mention this in the text.

Figure 2A. Please state clearly in the body of the manuscript that this is flow cytometry data. This was not obvious from reading the text.

We have clarified that this is flow cytometry data and added more information on the gating strategy and the new Figure 1—figure supplement 1: “We performed flow cytometry followed by t-distributed stochastic neighbor embedding (tSNE) analysis to evaluate immune cell heterogeneity and surface marker expression in the visceral adipose SVF (Figure 2A). tSNE analysis was performed based on gating strategies detailed in Figure 1—figure supplement 1.”

Figure 2E: Switch CD206 (main figure) and CD301 (supplementary).

We agree with the reviewers that the RELMalpha specific difference for CD301b may be of greater interest and have switched these figures.

Figure 2F: What is the significance of higher Siglec-F expression by eosinophils? please elaborate further on why this is important.

The original rationale behind investigating the MFI of different eosinophil markers was our observation in the tSNE that eosinophils were heterogeneous according to tSNE1 and tSNE2 between groups (see Figure 2B). We indeed saw differential expression of SiglecF, with female RELMalpha KO eosinophils expressing significantly higher SiglecF than WT Eos. Studies using mice deficient in SiglecF identified a role for SiglecF in inhibiting lung allergic responses (PMID: 17272508), but other studies show it promotes type 2 inflammation (PMID: 34996839; PMID: 27690378). The former study showed that SiglecF stimulation promotes apoptosis (PMID: 17272508). It is possible that the increased SiglecF expression in KO eosinophils can lead to increased apoptosis of these cells, thereby explaining the reduced eosinophil frequency, but we have no supportive data for this. Nonetheless, we provide these citations and discussion for the readers: “Siglec-F is a paralogue of human Siglec-8, and in mice is expressed on eosinophils but also alveolar macrophages. The function of Siglec-F appears to be context-dependent, with reported evidence of stimulatory and inhibitory roles on eosinophils (PMID: 34996839; PMID: 27690378). One study showed that SiglecF stimulation induced apoptosis (PMID: 17272508). It is possible that the increased expression of SiglecF on the RELMα-deficient eosinophils from HFD KO female mice may increase their susceptibility to apoptosis, explaining their reduced frequency”.

Line 206: It would be more proper to say M1-like macrophage, a classic M1 macrophage is defined by stimulation with LPS, and multiple papers over the years have shown CD11c+ macrophages in HFD models do not express classic M1 macrophage genes (PMID: 25242226).

We agree and have changed to M1-like throughout the text, where relevant.

Figure 4D-F does not seem to be representative flow plots based on the data graphed below. Recommend choosing a flow plot that represents the median of the graphed data.

We have changed the flow plots to show better representation of the median.

Consider merging figures 5 and 6, as they are both limited in scope.

We agree with the reviewer and have merged the two figures as suggested – see new Figure 5.

Figure 7a: Is the annotation able to differentiate fibroblast vs pre-adipocytes? Also in the Figure to the right of the TSNE plot, it is hard to determine the sex and genotype of these mice. Please draw groups on the x-axis like in 7C.

We thank the reviewers for this comment, since it would be significant to determine if RELMalpha, HFD, and/or sex influences adipocyte differentiation, which may contribute to the observed phenotype. We subclustered fibroblast population and analyzed the distribution of genes involved in adipocyte differentiation. Unfortunately, all of the genes were evenly distributed throughout this population that was designed as fibroblasts based on markers. Thus, we are unable to determine if one subpopulation of fibroblasts could be identified as pre-adipocytes. We mention this in the text.

Reviewer #3 (Recommendations for the authors):Below is a point-by-point critique to aid the authors in understanding specific points from the public review.Title – We recommend including "in young mice" as part of the title.

We agree that our studies are not investigating aged/old mice, so they could be considered ‘young’, however, the mice we utilize are sexually mature and 18-weeks old at the time of sacrifice. We do not think this would characterize them as ‘young’ from the perspective that this study is not looking at aging-induced inflammation. Therefore, we are concerned adding ‘young’ may detract from the main goals of the study. Instead, we add discussion on the age of the mice used so readers are aware.

Introduction – We believe that the introduction could benefit from editing down. Some of the details would be more appropriate for the discussion. Conclusions from the scRNA-Seq analyses will need to be updated after correcting the analyses.

Based on this suggestion, we removed all conclusions from the introduction. We now provide more information about the scRNA-seq analysis and all our validation steps. In addition, we perform new studies to investigate hemoglobin expression (see new Figure 7). Together, these expanded analyses and validation steps, and the new data, support our original conclusions, which we have kept in the text.

Results – An experimental design figure prior to Figure 1 and Figure 4 would be helpful to the reader.

We have now provided a new 4A panel detailing the experiment design. Due to space constraints for Figure 1, we have instead added an experimental design diagram in new Figure 1—figure supplement 1A.

There are several instances in the text where the authors claim that there is a significant difference between the two groups, but the statistics for these comparisons are not shown in the figure.

We have addressed all the specific comments below from the reviewer and ensured that our text concerning ‘significant differences’ reflects the figures.

– Line 97-98 – "Under both Ctr and HFD conditions, female mice had significantly higher RELMα in the serum and visceral adipose tissue than males (Figure 1A)." – It does not appear that there is a significant difference in RELMa within the female visceral fat under HFD.

We apologize for this error and have edited accordingly: “Under both Ctr and HFD conditions, female mice had significantly higher RELMα in the serum than males, and in adipose tissue under Ctr diet. Exposure to HFD diminished adipose RELMa levels in both sexes (Figure 1A).

– Figure 1B – Why was a t-test used for this figure when there were four groups to compare? Were any comparisons made between the Ctr groups? In the text (lines 100-101), it states "RELMα 101 deficiency did not affect Ctr or HFD weight gain in males…"; however these statistics are not described in the legend or text.

We agree that a t-test is not appropriate, so we have re-run the analysis as a two-way ANOVA (diet and genotype) with a post Sidak’s multiple comparison test. It is too complicated to show all the comparisons (hence we show analysis in 1C at specific timepoints with three-way ANOVA). Instead, we focus on significant differences between the WT and KO groups. We clarify in the figure legend that the comparison is between WT and KO mice (matching for diet). We didn’t see much difference with the control diet with either males or females but observed significant RELMalpha-specific differences with HFD in females but not males.

– Lines 107-108 – "The monocyte chemoattractant CCL2 was significantly elevated in HFD-males…" – Was this statistic assessed? It is not shown in Figure 1C.

We performed three-way ANOVA (diet, sex, genotype) with multiple comparison tests and there was no statistical difference between Ctr and HFD males likely due to the high variability, therefore we removed this statement from the text.

– Lines 111-12 – "The higher level of IL-10 and GM-CSF in females were dependent on RELMα and were further decreased with exposure to HFD…" – In figure 1E, the effect of HFD was not shown to be statistically significant.

We agree; by three way ANOVA (diet, sex, genotype) with multiple comparison tests, there was no statistical difference with HFD so we have removed this statement.

– Line 112 – "…while IL-5 was only decreased with HFD" – this statistic was not shown in Figure 1E.

We cannot perform statistics as the IL-5 was not detected (ND) with HFD, therefore we changed to “while IL-5 was not detected when mice were treated with HFD”

– Figure 2B – Quantify and show statistics for the claim on lines 140-141 "Ctr-fed male and female mice had high eosinophil numbers, and this subset disappeared in male mice upon HFD".

We now refer to Figure 1I, which quantified this.

– Figure 3E – It is difficult to determine which comparisons are being highlighted on the box plots. In the text (lines 154-156) the authors claim that this is an HFD effect – does this hold for both sexes?

The correlation analyses in Figures3E-F are showing that there is a significant negative correlation between weight and frequency of RELMalpha-expressing macrophages in HFD (Figure 3G) but not Ctr diet females (Figure 3F). Figure 3E combines both HFD and Ctr-diet female mice. We did not perform RELMalpha intracellular staining in males since our previous data shows the phenotype in females. Nonetheless, our single cell seq data and our IF staining does show RELMalpha expression in males. We clarify the text.

It is unfortunate that eosinophils could not be identified in the single-cell analysis since this population of cells was shown to be important in rescuing the RELMα-deficiency in HFD-fed females. The authors should note in the discussion how future scRNA-Seq experiments could overcome this limitation (i.e., enriching immune cells prior to scRNA-Seq). Although another scRNA-Seq experiment including eosinophils would add to the report, we believe it is beyond the scope of this study.

We were indeed disappointed that we were not able to obtain eosinophil single cell seq, but realize that this is a reported issue in the field. We have expanded our discussion of this (see response to Rev 3, comment #2 and to Rev 1, comment #3).

– Lines 283-286 – "We employed targeted approaches to identify eosinophil clusters according to eosinophil markers (e.g. Siglecf, Prg2, Ccr3, Il5r), and relaxed the scRNA-Seq cutoff analysis to include more cells and intronic content, but still could not find eosinophils." – Please provide the feature plots showing expression across clusters for Siglecf, Prg2, Ccr3, Il5r as supplemental figures.

We thank the reviewer for this suggestion and provided feature plots showing expression across all clusters In Figure 2—figure supplement 1G.

– Lines 286-288 – "We conclude that eosinophils may be absent due to the enzyme digestion required for SVF isolation and processing for single-cell sequencing, which could lead to specific eosinophil population loss due to RNases or cell viability issues." – Do you use the same single cell dissociation techniques for flow cytometry as were employed for the scRNA-seq? If so, what proportion of cells are eosinophils?

The reviewer is correct that the same tissue dissociation approach was employed for scRNA-seq and flow analyses, but the differences arose in that flow analyses identified populations based on protein, while scRNA-seq uses RNA expression of cell-specific genes. As described above, granulocytes express cell surface protein markers such as SiglecF used to identify this population in flow cytometry and calculate the proportion presented in Figure 1I; but are very RNA-poor as described before and degranulation can lead to further RNA degradation during tissue processing. This has been clarified in the methods and the discussion.

There is inconsistency in the presentation and interpretation of the correlations in Figure 3.

We have edited the text to respond to these reviewer comments, below.

– The correlations in Figure 3A, 3B, and 3E appear to be driven entirely by diet effect. Is it possible that the HFD and not body weight are responsible for this effect?

We agree that this is a possibility, and have added this in the text.

– Figure 3C-G – Why are males absent from this analysis?

Intracellular staining for RELMalpha in the visceral adipose tissue cells is a long process and not easily performed with many animals. We initially did surface staining for all the groups (sex, genotype, diet), but when we saw a difference in the females, we focused on females for the longer process of intracellular staining. We did not perform RELMa intracellular staining on males, however, our adipose tissue RELMalpha ELISA, our single cell sequencing, and our IF staining can provide some answers for the males.

– Figure 3E – Why are all groups not included in this particular correlation, as in Figure 3A and Figure 3B? Is this a female-specific effect? I understand why the KO groups were not included, but the text does not describe the sex difference here.

We cannot answer this question because we focused on females and did not perform intracellular cytokine staining on males. We feel that this is beyond the scope of this project, and have provided other measures for RELMalpha expression: ELISA, single cell sequencing, IF staining that includes the male groups.

– Lines 183-184 – "suggesting that the protective effect of RELMα may be impaired with body weight gain (Figure 3F-G)." – This result could also imply that HFD (and not body weight) is causing a decrease in RELMα. Do you think that this same effect would be seen in another obesogenic diet (i.e., a high carbohydrate diet)? We are not recommending adding these experiments to the current report, but authors may choose to comment on this in the discussion.

We agree with the reviewer that several studies have demonstrated that HFD per se, prior to weight gain can cause changes in immune cells. Most studies using diet-induced obesity models, are confounded with the combination of diet and obesity: animals are analyzed after they gain weight in response to HFD. This also creates problems delineating causes of downstream pathology, since obesity is a state of hyperleptinemia, hyperinsulinemia, hyperglycemia and hyperlipidemia, in addition to chronic inflammation. Several studies attempted to answer this question. For example, it was shown that HFD causes inflammation in the hypothalamus before the onset of obesity; only 1 day of HFD exposure caused cytokine induction (PMID: 16002529; PMID: 22201683). Another study demonstrated that HFD itself caused inflammation rather than obesity per se, using monogenic obese mice (ob/ob or db/db that are obese without exposure to HFD), at least with respect to brain macrophage /microglia activation (PMID: 24166765). On the other hand, later studies demonstrated that ob/ob mice also experience inflammation, demonstrated by cytokine increase, indicating that genetic obesity without HFD can contribute to inflammation (PMID: 32296847; PMID: 32422516).

We first analyzed the levels of RELMa after HFD and as demonstrated in Figure 1A and 1B, serum levels of RELMa are not affected by diet, but adipose tissue levels are, which may imply that increased tissue inflammation lowers RELMa. Figure 3F-G demonstrate correlation between weight and RELMa after exposure to HFD. Protective effects of RELMa were demonstrated in Figure 4 with RELMa treatment. The text has been revised to improve clarity. We consider that the same effect would be observed with any obesogenic diet, since increased adiposity causes inflammation, and RELMa decreases in inflammatory conditions, as a marker of anti-inflammatory myeloid cells. We have added discussion of this point.

There are several issues with the scRNA-Seq analysis and interpretation. More details on the steps taken in the single-cell analyses should be included in the methods section.– Deposition of code to a public repository (i.e., GitHub) would allow for rigor and reproducibility of the analyses.

We thank the reviewer for their suggestion and we uploaded all necessary code to perform these analyses on GitHub repository: https://github.com/rrugg002/Sexual-dimorphism-in-obesity-is-governed-by-RELM-regulation-of-adipose-macrophages-and-eosinophils/blob/main/Figure8.R.

With regards to the 'pseudobulk' analyses presented in Figures 5-6, several of the differentially expressed genes identified in Figure 6 are hemoglobin genes (i.e., Hba, Hbb genes). It is not uncommon to filter these genes out of single-cell analysis since their presence usually indicates red blood cell (RBC) contamination (PMID: 31942070, PMID: 35672358). We would recommend assessing RBC contamination as well as removing Figure 6 from the manuscript and focusing on cell-type-specific analyses.

Prior to our first submission, we consulted with 10x support scientists and the UCR bioinformatics core director to ensure that our analyses included the appropriate filtering. We have now detailed this in the methods, and believe our analysis is robust (and this is confirmed by the experts we consulted above). The PMIDs provided above are from studies that looked at hippocampus development (where they didn’t perfuse so there may be blood contamination) or whole blood (where there would be significant red blood cell contamination). Instead, we perfused our mice and treated the single cell suspension with RBC lysis buffer, see methods. Also, we have now extended our scSeq analysis to compare hemoglobin RNA to red blood cell specific marker Gypa/CD235a. While hemoglobin is distributed throughout the myeloid population in the female KO mice, Gypa/CD235a, which would suggest RBC contamination is not expressed at all (see new Figure 7). Additionally, we provide hemoglobin protein ELISA and IF staining to support our finding that macrophages from KO mice express hemoglobin protein. Last, two publications support hemoglobin expression by nonerythroid sources, including macrophages (PMID: 10359765; PMID: 25431740). While we are confident based on above that our data is not due to RBC contamination, we cannot exclude the fact that, although unlikely, macrophages may be phagocytosing RBC and preserving specifically hemoglobin RNA and protein. Nonetheless, we discuss this possibility in the text. In conclusion, based on the justification above and the new data, we are confident that our findings and overall conclusions are robust.

To assess for potential RBC contamination, we looked at top marker genes expressed by murine erythrocytes (PMID: 24637361). We had a small cluster of potential RBCs (about 75 cells) that were filtered out prior to downstream DEG analysis. Please see Author response image 1 feature plots:

**Author response image 1. sa2fig1:** 

– Figure 5D – It is unclear what comparisons are generating these lists of genes. Please provide a supplemental table with all differentially expressed genes from the analyses.

We thank the reviewer for their suggestion. Figure 5D highlights the number of genes that are expressed uniquely in all four groups. We agree that this figure isn’t very clear, so we removed it as recommended and placed this figure in Supplemental Figure 3.

Within the text, there are several instances where the authors claim that a pathway is upregulated based on their Gene Ontology (GO) over-representation analysis (ORA). To come to this conclusion, the authors identify genes that are upregulated in one condition and then perform GO-ORA on these genes. However, the authors do not consider negative regulators, whose upregulation would actually decrease the pathway. Authors should either replace their GO-ORA analysis with one that considers the magnitude and direction of differentially expressed genes and provides an activation z-score (i.e., Ingenuity Pathway Analysis) or replace instances of 'upregulated' or 'downregulated' pathways with 'over-represented' pathways.

We thank the reviewer for their suggestion and we corrected all GO plots from “upregulated” or downregulated” to “over-represented pathway.”

For Figure 7A, a representative tSNE plot for each group (WT Female, KO Female, WT Male, KO Male) should be shown to ensure there is proper integration of the clusters across groups. There are some instances where the scRNA-Seq data do not appear to be integrated properly (i.e., Supplemental Figure 2C). The authors should explore integration techniques (i.e., Seurat; PMID: 29608179) to correct for potential batch effects within the analysis.

We thank the reviewer for the suggestion of proper integration of the clusters across groups. We performed integration using the Cell Ranger aggregation (aggr) pipeline (see updated Materials and methods section). In addition, many technical controls were performed to prevent batch effects between our samples. For sequencing, we used the 10x genomics library sequencing depth and run parameters for both gene expression and multiplexing libraries. For all 3’ gene expression library sequencing, we sequenced at a depth of 20,000 read pairs per cell and for all cell multiplexing library sequencing we sequenced at a depth of 5,000 read pairs per cell. All libraries were paired-end dual indexed libraries and were pooled on one flow cell lane using a 4:1 ratio (3’ Gene expression: Multiplexing ratio) in the Novaseq, as recommended by 10x Genomics, in order to maintain nucleotide diversity and prevent batch effects during the sequencing process. When performing integration/aggregation of all sample gene expression libraries using the Cell Ranger aggregation (aggr) pipeline, we performed sequencing depth normalization between all samples. Cell Ranger does this by equalizing the average read depth per cell between groups before merging all sample libraries and counts together. This is a default setting in the Cell Ranger aggr pipeline, and this approach avoids artifacts that may be introduced due to differences in sequencing depth. Thus, we are confident that changes we observed in gene expression and cell type populations are due to biological differences and not technical variability. In Author response image 2 we have provided a tSNE plot showing clustering of all 12 samples after we performed integration:

We updated Figure 7 and included a representative tSNE plot for each group. We also updated the tSNE plot for Supplemental Figure 4C (was previously S2C) showing overall clustering amongst all groups. The largest population differences occurred in the fibroblast population and these population differences were largely due to sex differences. Because we are confident that integration was performed appropriately and that batch effects were controlled for, we believe these sex differences are a biological effect.

– Lines 338-339 – "When analyzing scRNA-seq data from WT mice, RELMα (Retnla) is expressed in the Mono and Mac1 clusters (Figure 7D, F)." – It appears that RELMα is also expressed in the Mac4 cluster. Did you perform a plot to see if RELMα is present in any other clusters from Figure 7A?

We thank the reviewer for pointing this out. Retnla is expressed at the low level in all SVF clusters (now labelled Figure 6), but much higher in the myeloid cluster. This is not surprising, as we mention in the introduction and discussion. However, RELMa expression levels are much higher in monocyte-derived macrophages as the heatmap supports. Retnla is also expressed in some cells in Mac4, but this is much less than the Mono and Mac 1 clusters (see heatmap 6F). Therefore, for simplicity, we decided to focus on Mono and Mac 1 clusters.

LncRNA Gm47283 is identified as a gene that is differentially expressed by genotype in HFD females (Figure 7G); however, according to Ensembl this gene is encoded on the Y-chromosome (https://uswest.ensembl.org/Mus_musculus/Gene/Summary?g=ENSMUSG00000096768;r=Y:90796007-90827734). The authors should use the RELMα genotype and sex chromosomally-encoded genes to confirm that their multiplexing was appropriate.

We agree with the reviewer that it is crucial to confirm that multiplexing and all subsequent analyses are performed correctly. Comparison between males and females contains internal controls that increase confidence, such as Xist gene that is expressed only in females, and Ddx3y that is located on the Y chromosome. LncRNA, Gm47283 is located in the syntenic region of Y chromosome and is also annotated as Gm21887 located in the syntenic region of the X chromosome. It also has 100% alignment with Gm55594 on X chromosome, following BLAST analysis. Additionally, it is also referred to as erythroid differentiation regulator 1 (Erd1), x or y depending on the chromosome, although NCBI database specifies partial assembly and incomplete annotation. We revised the manuscript and referred to this lncRNA as Gm47283/Gm21887 to prevent further confusion.

For Figure 8, samples should be co-clustered and integrated across groups before performing trajectory analysis to allow for direct comparisons between groups.

We appreciate the valuable feedback and suggestions, which have been helpful in clarifying the trajectory analysis, which we have done as follows:

Regarding the co-clustering and integration of our samples across groups, here is the explanation of our trajectory analysis approach. We have co-clustered all of our samples using the align_cds function from the Monocle3 package. We have included the code for Figure 8 in our GitHub repository at https://github.com/xqiu625/Sexual-dimorphism-in-obesity-is-governed-by-RELM-regulation-of-adipose-macrophages-and-eosinophils/blob/main/Figure8.R. Specifically, code lines 138, 166, 196 and 225 indicate that we have used the align_cds function to cluster our samples by "Sample.ID".

The align_cds function in Monocle3 can be used to co-cluster all samples in a single-cell RNA-seq experiment by aligning coding sequences (CDS) across different cell types or conditions. The align_cds function takes a set of reference CDS sequences and single-cell RNA-seq reads and identifies the CDS sequences within each read, allowing the identification of differentially expressed genes across different cell types or conditions based on the aligned CDS sequences. More details about align_cds can be found here https://rdrr.io/github/cole-trapnell-lab/monocle3/man/align_cds.html.

We hope that this additional information will be helpful in further understanding our analysis approach.

– Line 289-291 – "The main population in the SVF, accounting for 50-75% of cells, were non-immune cells identified as Pdgrfa+ fibroblasts (green). These were significantly increased in WT females compared to the other groups (p<0.01, S2C)." – Were these data integrated using a feature like the "IntegrateData" command in Seurat? It does not appear that the cells cluster well when comparing males and females.

The data were Integrated using the Cell Ranger aggr pipeline. Please see above comment (#24) for more details regarding integration and sequencing depth normalization. We have also included these details in our updated Materials and methods section.

Scale bars on several heatmaps are not labeled with what they represent (Figures 5E, 7F, 7G, S2N, S2O, S4B, S4C).

We thank the reviewer for noticing these errors and we labeled the scale bars on all the above figures with what they represent.

Since the experiments presented in this report were from young mice using a single diet intervention, the authors should comment on how age and other obesogenic diets may impact the results found here. Also, the authors should expand their discussion as to what upstream regulators (i.e., hormones or genetics) may be driving the sex differences in RELMα expression in response to HFD.

Please see the response to comment #8. Extensive discussion on all possible reasons for sex differences would be challenging in the limited space. To address the reviewer’s comments, we focus on several possibilities such as estrogen, intrinsic differences in immune system and sex differences in adipokine levels and metabolism.

“Several studies addressed protective role of estrogen in obesity-mediated inflammation and weight gain, which may occur via estrogen regulation of RELMa levels, and is a focus of our future studies. Alternatively, intrinsic sex differences in immune system have been demonstrated as well (30254630, 33268480, 25869128) that are dependent on sex chromosome complement and/or Xist expression (34103397, 30671059), and RELMa may be regulated by these as well. Additionally, ageing-mediated increase in inflammation (including of adipose tissue, recently reviewed in 36875140), may also occur via changes RELMa levels and will be addressed in future studies.”